



# Rates and drivers of Red Sea plankton community metabolism

Daffne C. López-Sandoval[1], Katherine Rowe[1], Paloma Carillo-de-Albonoz[1], Carlos M. Duarte[1] and

Susana Agusti[1]

[1] Red Sea Research Center, King Abdullah University of Science and Technology (KAUST), Thuwal-Jeddah, 23955-6900,

Saudi Arabia

*Correspondence to*: Daffne C. López-Sandoval (daffne.lopezsandoval@kaust.edu.sa)

## Abstract

Resolving the environmental drivers shaping planktonic communities is fundamental to understanding

their variability, present and future, across the ocean. More specifically, resolving the temperature-

dependence of planktonic communities in low productive waters is essential to predict the response of

marine ecosystems to warming scenarios, as ocean warming leads to oligotrophication of the

subtropical ocean. Here we quantified plankton metabolic rates along the Red Sea, a unique

oligotrophic and warm environment, and analysed the drivers that regulate gross primary production

(GPP), community respiration (CR) and the net community production (NCP). The study was

conducted on six oceanographic surveys following a north-south transect along Saudi Arabian coasts.

Our findings revealed that Chl-*a* specific GPP and CR rates increased with increasing temperature ($R^2$ =

0.41 and 0.19, respectively, P< 0.001 in both cases), with a higher activation energy (AE) for GPP (1.2

± 0.17 eV) than for CR (0.73 ± 0.17 eV). The higher AE for GPP than for CR resulted in a positive

relationship between NCP and temperature. This unusual relationship is likely driven by 1) the

relatively higher nutrient availability found towards the warmer region (the South of the Red Sea), and

which favours GPP rates above the threshold that separates autotrophic from heterotrophic communities

(1.7 mmol $O_2$ m$^{-3}$ d$^{-1}$). 2) Due to the arid nature, the basin lacks riverine and terrestrial inputs of organic

carbon to subsidise a higher metabolic response of heterotrophic communities, thus constraining CR



rates. Our study demonstrates that GPP increases steeply with increasing temperature in the warm ocean when relatively high nutrient inputs are present.

## 1 Introduction

Plankton community metabolism, the balance between gross primary production (GPP) and

respiration (CR), defines the carbon available to fuel pelagic food webs and determines whether plankton communities act as a source or sink of $CO_2$ (Duarte and Agustí, 1998; Williams, 1998; Duarte et al., 2011). Whereas GPP typically satisfies the respiratory demands within the food web across productive waters, the oligotrophic ocean often requires allochthonous inputs of organic carbon to meet the metabolic requirements of heterotrophic organisms (Duarte et al., 2013). Due to comparatively

higher carbon consumption, relative to the production (GPP), planktonic communities in low productive systems are in close metabolic balance (i.e., net community metabolism, NCP = 0, or GPP = CR) or experience a net metabolic imbalance (i.e. NCP < 0, GPP < CR) (Smith and Hollibaugh, 1993; Duarte and Agustí, 1998; Duarte et al., 2013).

In tropical and subtropical oligotrophic regions, the high temperatures, which are a characteristic

of these systems, may amplify the metabolic imbalances in plankton communities, as CR tends to increase faster than GPP with increasing temperature (Harris et al., 2006; Regaudie-de-Gioux and Duarte, 2012) if allochthonous sources of organic carbon subsidize their carbon demand. These allochthonous inputs may be delivered from land through riverine discharge, from the atmosphere through atmospheric deposition of dust and volatile organic carbon (Jurado et al., 2008), or exported

from productive coastal habitats (Duarte et al., 2013; Barrón and Duarte, 2015).



The Red Sea is semi-enclosed highly oligotrophic basin (Acker et al., 2008; Raitsos et al., 2013), known as one of the warmest tropical seas, with maximum sea surface temperatures ranging from 33.0 to 33.9 ℃ during the summer period (Chaidez et al., 2017; Osman et al., 2018). Although in specific regions, temperatures can reach 34 - 35 ℃ (Rasul and Stewart, 2015; Garcias-Bonet and Duarte, 2017;

Almahasheer et al., 2018). As a result to its desert conditions, the Red Sea experiences large evaporation rates (nearly 2 cm yr$^{-1}$ of freshwater from the surface layers), while the lack of river runoff and low precipitation rates make this system one of the saltiest seas on the planet (Sofianos, 2002; Sofianos and Johns, 2015; Zarokanellos et al., 2017). Two wind patterns govern the region: In the northern part, the wind coming from the northwest remains relatively constant through the year, while

in the southern area, the Indian Monsoon system regulates the wind dynamics. (Sofianos, 2002; Sofianos and Johns, 2015). During the winter monsoon, the wind changes direction, and this wind reversal along with the thermohaline forces drives the overall circulation and favours the exchange of water with the Indian Ocean (Sofianos, 2002; Zarokanellos et al., 2017).

Due to the almost negligible terrestrial inputs, the intrusion of nutrient-rich waters from the

Indian Ocean through the Bab-el-Mandeb Strait (Sofianos and Johns, 2007; Raitsos et al., 2015; Kürten et al., 2016), together with aeolian dust and aerosol deposition (Chen et al., 2007; Engelbrecht et al., 2017), represent some of the primary sources of nutrients into the basin. Thus, nutrient availability in the Red Sea follows a latitudinal pattern that is opposite to the one of salinity, but parallel to the thermal gradient, with nutrient-richer and warmer waters towards the Southern Red Sea compared to the cooler

and more oligotrophic Northern Red Sea (Sofianos, 2002; Raitsos et al., 2015).



Studies based on ocean color data revealed that chlorophyll-*a* (Chl-*a*) concentrations decline

from the Southern Red Sea to the Northern Red Sea (Raitsos et al., 2013; Kheireddine et al., 2017;

Qurban et al., 2017) and depict a clear seasonality. During winter time, when the maximum exchange of

water with the Indian Ocean takes place, Chl-*a* concentration peaks, decreasing towards the summer

period when the water column is mostly stratified (Sofianos, 2002). Measurements of primary

production also revealed that phytoplankton photosynthetic rates follow the same south to north

gradient as Chl-*a* and nutrient concentration (Qurban et al., 2017). However, there is no information

regarding the metabolism of the plankton communities.

Based on available evidence, we hypothesise that: (1) the high gross primary production

expected in the Southern Red Sea may be counterbalanced by a higher respiratory demand in these

warm waters (Harris et al., 2006; Regaudie-de-Gioux and Duarte, 2012), and that (2) NCP might

decline towards the unproductive waters of the Northern Red Sea. With the expected decrease in GPP

towards the northern region, planktonic metabolism might be driven mainly by heterotrophic

communities (Duarte and Agustí, 1998; Duarte et al., 2013). However, the absence of significant

allochthonous subsidies in the basin may hamper the metabolic response of the heterotrophic plankton

communities. Hence, it remains unclear what the metabolic balance of plankton communities is and

whether south to north gradient in NCP exists in the Red Sea.

Here we report the variability of plankton community metabolism (GPP, CR and NCP) along the

Red Sea, and examine if the temperature-dependence of planktonic metabolic rates in this basin are

consistent with those reported for the global ocean (López-Urrutia et al., 2006; Regaudie-de-Gioux and

Duarte, 2013; Garcia-Corral et al., 2017). We did so by measurements conducted in six surveys along



the south-north latitudinal gradient in the Saudi Economic Exclusive Zone in Red Sea waters. We

determined plankton metabolic rates at three different optical depths between winter 2016 and spring

2018, thus allowing us to 1) delineate the seasonal variability of the gross primary production (GPP)

and community respiration (CR) along the Red Sea, 2) quantify changes in the metabolic balance (net

5 community production, NCP) and 3) test the hypothesized roles of productivity gradients and

temperature in driving NCP.



## 2. Material and Methods

### 2.1 Field Sampling

We conducted six oceanographic surveys [two during autumn (October and November 2016), two during winter (February 2016 and January 2017), one in summer (August 2017), and one in spring

(March 2018)] on board the R/V *Thuwal* and R/V *Al Azizi*. Sampling was conducted following a latitudinal transect along the Red Sea within a region limited by coordinates 17.25 °N to 27.82 °N and 34.83 °E to 41.39 °E (Fig. 1), sampling between five and seven stations per survey. At each station, vertical profiles of temperature and salinity were obtained with a Sea-Bird SBE 911 plus CTD profiler (Sea-Bird Electronics, Bellvue, WA, USA), with additional sensors to measure the attenuation of

photosynthetically active radiation (PAR) (Biospherical/Licor PAR/Irradiance Sensor), *in vivo* fluorescence (WetLabs ECO FL fluorometer), and dissolved oxygen concentration (Seabird SBE 43 Dissolved Oxygen Sensor). Water samples for chemical and biological measurements were collected between 7:00 and 9:00 am local time, using a rosette sampler equipped with 12 Teflon Niskin bottles (12 L) that were provided with silicone O-rings and seals.

### 2.2 Inorganic nutrients and chlorophyll-*a* concentration

Water samples for nutrients analysis were collected in 50 mL polyethene bottles and kept frozen (-20 ℃) until determination. Inorganic nutrient concentration was determined with a SEAL AA3 Segmented Flow Analyzer (SEAL Analytical Inc., WI, USA) using standard methods (Hansen and Koroleff, 1999). The detection limits were 0.05 µM for nitrate, 0.01 µM for nitrite, 0.01 µM for



phosphate and 0.08 μM for silicate. For the chlorophyll-*a* analysis (Chl-*a*), 200 mL samples were taken

at ten discrete depths (between 5 and 200 m) and filtered through Whatman GF/F filters. The filters

were kept frozen until analysis (-20 ºC). Pigments were extracted using 90 % acetone for 24 h and left

overnight in the dark at 4 ºC. Chl-*a* concentration was estimated with the non-acidification technique

using a Trilogy Fluorometer equipped with CHL-NA module (Turner Designs, San Jose, USA),

previously calibrated with pure Chl-*a.*

## 2.3 Net community metabolism, community respiration and gross primary production

Planktonic metabolic rates were determined by changes in dissolved oxygen concentration

during 24 h incubations (Carpenter, 1965). We selected three different optical depths, one of them at a

depth of the chlorophyll maximum (i.e. 100 %, 60–20 % and 8–1 % of PAR). Seawater was collected

directly from the Niskin bottles to fill a total of 21 (100 mL) Winkler bottles. The bottles were carefully

filled using silicone tubing and allowing the water to overflow during the filling, taking special care to

avoid the formation of air bubbles. Surface samples (100 % PAR) were collected in 100 mL quartz

bottles. From each depth, seven of the bottles were immediately fixed with Manganese sulfate ($MnSO_4$)

and Potassium hydroxide/Potassium iodide solution (KI/KOH) to determine the initial oxygen

concentration while the other 14, seven light and seven black bottles, were incubated on deck in surface

water flow through tanks. Due to the difference in temperature between the surface and deep waters,

particularly during the summer and autumn surveys, we decided to include in our analyses only those

samples collected above the thermocline. Changes in temperature and PAR in the incubation tanks was

recorded with HOBO Pendant data loggers (Onset, Massachusetts, USA).



Before the incubation, the bottles were covered with neutral mesh to reduce the incident PAR

radiation according to the sampled depth. At the end of the incubation period, light and dark bottles

from each depth were fixed to determine final $O_2$ concentrations. Oxygen concentration was measured

by automated high-precision Winkler titration with a potentiometric end-point detection (Oudot et al.,

1988) using a Mettler Toledo T50 Titration Excellence auto-titrator attached to an Inmotion

autosampler. Net community metabolism (NCP, mmol $O_2$ m$^{-3}$ d$^{-1}$) was calculated as the difference in

the oxygen concentration between the light bottles after the 24 h incubation period and the oxygen

concentration measured before the incubation. Community respiration rates (CR, mmol $O_2$ m$^{-3}$ d$^{-1}$) were

calculated as the difference of the oxygen concentration after the 24 h incubation period in the dark

bottles and the initial oxygen concentration. Gross primary production (GPP, mmol $O_2$ m$^{-3}$ d$^{-1}$) was

calculated as the sum of NCP and CR.

Due to the consistent relationship existing between plankton metabolism and temperature across

diverse marine regions (Regaudie-de-Gioux and Duarte, 2012; García-Corral et al., 2014), we examined

how plankton metabolic rates covariate with temperature in the Red Sea, a system whose temperature

range is higher than previously encountered in marine planktonic metabolism research. We determined

the relationship between metabolic rates and temperature by fitting an OLS (Ordinary least squares)

linear regression equation to the relationship between the natural logarithm of the Chl-*a* specific

metabolic rates and the inverse of the absolute temperature * *k*, which is the Boltzmann's constant

(8.617734 *$10^{-5}$ eV K$^{-1}$). In these Arrhenius plots, the slope represents the average activation energy

(AE), characterising the extent of thermal-dependence of metabolic processes.



## 2.4 Statistics

Statistical analysis was done using GraphPad Prism 7 (GraphPad Software, La Jolla, CA, USA) and MVAapp_v2.0 (Julkowska et al., MVApp.pre-release_v2.0 mmjulkowska/MVApp: MVApp.pre-release_v2.0. DOI: 10.5281/zenodo.1067974 ).

## 3. Results

### 3.1 Latitudinal variability of physico-chemical properties and Chl-*a* concentration

The temperature and salinity profiles revealed strong latitudinal and seasonal variation in the structure of the water column along the basin (Figures 2 and A1). During the summer, sea surface temperature (SST), ranged from 32.5 ℃, at the southern-most station (17 ℃N), to 29.3 ℃ at the far-northern sampling site (27 ℃N) (Figure 2). During this survey, the concentration of dissolved inorganic nitrogen (DIN, $NO_2 + NO_3$) decreased from 0.61 μM (± 0.11) (at the first optical depth) to 0.12 μM (± 0.04) between 15–5 % of PAR, while phytoplankton chlorophyll-*a* concentration remained ∼ 0.2 μg Chl-*a* l$^{-1}$ (Table 1). In Autumn, SST ranged from 31.2 ℃ (southern-most station) to 27.4 ℃ (northern-most station) (Figure 2), while nutrient availability and Chl-*a* concentration increased, particularly at the southern stations where phytoplankton Chl-*a* concentration was above 0.8 μg l$^{-1}$ in the surface layers (Figure 2, Table 1). During winter, the water column generally remained well mixed, with temperatures fairly similar along the upper 100 m (Figure A1). During this period, Chl-*a* concentration along the basin peaked (Figure 2) while DIN, phosphate and silicate concentration remained similar within the first two optical depths (Table 1).

We identified two distinctive patterns portraying nutrient distribution along the Red Sea.

Nutrient concentration throughout the water column decreased with latitude (Figure 3), while DIN

concentration increased with temperature in the first optical depth (37 % PAR) and at the base of the

euphotic zone (1–0.1 % PAR) (Figure 3A and 3B). Phosphate and silicate concentrations were also

positive correlated with temperature at the base of the photic layer (Pearson correlation, DIN, r = 0.39, P

= 0.01; phosphate, r = 0.48, P < 0.001; silicate, r = 0.42, P< 0.001) (Figure 3B). Below the first optical

depth and above the base of the euphotic zone, nutrient availability and temperature were not correlated

(data not shown). In general, our results confirm that all variables correlated significantly with latitude,

highlighting the prevalence of the south to north gradient in temperature, salinity, nutrient availability

and chlorophyll-*a* concentration across the Red Sea.

### 3.2 Variability of plankton metabolism measured along the Red Sea

The measurements of plankton metabolic rates taken from the six oceanographic surveys

(between winter 2016 and spring 2018) allowed us to define the general variability patterns of gross

primary production (GPP) and community respiration (CR) along the Red Sea (Figures 4 and 5, Table

2). Plankton communities were autotrophic when all surveys were taken in concert, with heterotrophic

communities representing 38 %, 32 % and 56 % of the communities assessed between 100–37, 36–6,

and 5–1 % PAR, respectively (Table 2). Part of the variability in community metabolism was explained

by seasonal differences, as plankton communities tended to be mostly heterotrophic (80 % of NCP < 0)

along the basin during spring, while between summer and winter, NCP rates < 0 were mostly restricted



to the northern part of the Red Sea (above 21 ºN) (Figure 6). GPP (and CR) rates peaked in autumn and

winter (3–8 mmol $O_2$ $m^{-3}$ $d^{-1}$) in the southern stations, while in the stations sampled towards the north

GPP rates remained ~ 1.5 mmol $O_2$ $m^{-3}$ $d^{-1}$ throughout the year (Table 2, Figure 4). Plankton metabolic

rates were independent of dissolved inorganic nitrogen concentration ($P = 0.99$, 0.47, and 0.43 for GPP,

CR, and NCP, respectively), but increased significantly with increasing Chl-*a* concentration ($R^2_{(GPP)} = $

0.50, $R^2_{(CR)} = 0.45$ and $R^2_{(NCP)} = 0.14$, $P < 0.001$ in all cases) and temperature ($R^2_{(GPP)} = 0.23$, $R^2_{(CR)} = $

0.15, $R^2_{(NCP)} = 0.11$, $P < 0.001$ in all cases) (Table 3, Figure 8).

      When we evaluated the relationship of GPP with CR and NCP (Figure 7A and B), the analysis

showed that CR and NCP increased significantly with GPP ($R^2 = 0.62$ and 0.50, respectively; P

<0.001). From the functional relationships between GPP with CR and NCP, we calculated the threshold

of GPP for metabolic equilibrium for the region. By solving the fitted linear regressions using the slope

and intercept obtained from the OLS analysis (Figures 7A and B) for GPP, where GPP = CR and GPP

where NCP = 0, we determined that the GPP threshold that separates autotrophic from heterotrophic

planktonic communities in the Red Sea is 1.7 mmol $O_2$ $m^{-3}$ $d^{-1}$ (range 1.2–1.9 mmol $O_2$ $m^{-3}$ $d^{-1}$).

**3.3 Metabolic rates and temperature**

      We further explored the temperature-dependence of GPP and CR, by analysing the relationship

between the normalised metabolic rates (Chlorophyll-*a* specific metabolic rates) and temperature. This

analysis showed that both GPP and CR tended to increase with temperature albeit with different

activation energies (AE) (Figure 9). The AE was significantly higher for GPP (-1.2 ± 0.2 eV) than for

CR rates ( -0.73 ± 0.2 eV). We also tested whether the temperature-dependence response observed in



GPP and CR varied between seasons (Figure 10). The results revealed that during summer, the AE for

GPP rates was higher (-2.32 ± 0.76 eV) than for the rest of the seasons (values ranged between 1.5–1.8

eV); although the seasonal activation energies were not significantly different (F = 0.39, dF= 3, P=

0.76). The metabolic energy related to respiratory processes of the planktonic community (CR) change

significantly between seasons (F = 5.25, dF = 3, P = 0.002), with the highest values found in spring (-

2.63 ± 0.9 eV), and the AE for CR decreasing to -1.4 eV during summer and autumn (Figure 10B).

During winter, the correlation between Chl-specific CR rates and 1/ $k$T was not statistically significant

(r = -0.04, P = 0.89).



## 4. Discussion

### 4.1 Variability of plankton community metabolic rates along the Red Sea

Our results demonstrate that planktonic metabolic rates are markedly different between the southern (below 21–22 ºN) and the northern half of the Red Sea basin (above 21–22 ºN). Gross primary production and community respiration rates varied, on average, by a factor of 3–5 between the northern and the southern regions (during the autumn and winter period). The metabolic balance (NCP) showed that planktonic communities tended towards heterotrophy in the northern region (particularly during the summer and winter period), whereas the southern Red Sea tended to be mostly autotrophic, except in spring, when all the stations < 22 ºN were heterotrophic. Consistent with these findings, our data revealed that the GPP threshold that separated autotrophic from heterotrophic communities in the Red Sea was 1.7 mmol $O_2$ $m^{-3}$ $d^{-1}$, similar to that reported across oceanic communities elsewhere (Duarte and Agustí, 1998; Duarte and Regaudie-de-Gioux, 2009).

The low productivity of the Northern section of the Red Sea (above 21–22 ºN), with average GPP of $1.57 \pm 0.16$ mmol $O_2$ $m^{-3}$ d-1, explains the prevalence of heterotrophic communities therein. Sustaining heterotrophy in oligotrophic regions requires an allochthonous source of organic matter (Duarte et al. 2011, 2013). The arid nature of the northern Red Sea, with the watershed consisting mostly of deserts, leads to the absence of rivers and significant organic carbon inputs to the sea. Dust inputs are important, however, and whereas they have shown no effect on primary production (Torfstein and Kienast, 2018), they are a source of organic carbon (Jurado et al. 2009) that can partially supply the carbon required to sustain heterotrophic communities. Moreover, the Red Sea contains highly





productive coral reef, mangrove, seagrass and algal communities in the extensive shallow coastal areas (Rasul et al., 2015; Almahasheer et al., 2016) which may export significant organic carbon to the pelagic compartment, thereby helping to sustain heterotrophic plankton communities in the Northern Red Sea.

5       The significant correlations observed between metabolic rates with Chl-*a* concentrations (positive relationship), and with latitude (inverse correlation) further supports the idea that different ecological domains govern planktonic metabolism in the Red Sea. In the southern half (below 21–22 ºN), the balance between GPP and CR generally resulted in positive NCP values, i.e. autotrophic communities, while in the northern half (above 21–22 ºN) the metabolic balance resulted in NCP values

< = 0, i.e. heterotrophic communities. The latitudinal differences depicted in our results mirror the pattern of increasing Chl-*a* concentration and primary production towards the south of the Red Sea (Acker et al., 2008; Raitsos et al., 2013; Qurban et al., 2014; Kheireddine et al., 2017). Additionally, those differences are visible in the presence and distribution of distinctive planktonic communities between the north and south (Al-aidaroos et al., 2016; Pearman et al., 2016; Robitzch et al., 2016;

Kheireddine et al., 2017; Kottuparambil and Agusti, 2018).

## 4.2 Temperature and metabolic balance in the Red Sea

      Temperature is a master variable that regulates many components of ocean dynamics, such as vertical stratification, and most aspects of organismal biology, from setting boundaries in the distribution of organisms (Clarke, 1996) to controlling biochemical reactions that constrain the energy

for metabolic processes (Gillooly et al., 2001). Hence, the temperature is likely a significant driver of



metabolic processes in the Red Sea, one of the warmest tropical marine ecosystems (Raitsos et al.,

2011; Chaidez et al., 2017). Indeed, our results showed a positive response of planktonic metabolism to

temperature. Moreover, the functional relationships between the chlorophyll-*a* specific metabolic rates

with temperature suggested that both GPP and CR were positively enhanced with increasing

temperature; but at a different pace.

The metabolic theory of ecology (MTE), that relates the metabolic rate of an organism with its

mass and temperature, hypothesizes that individual metabolic rates vary with temperature with an

activation energy (minimum kinetic energy for the molecules to react, AE) relatively constant (AE ~

0.63 eV) from unicellular organisms to plants and animals (Gillooly et al., 2001; Brown et al., 2004).

For aerobic respiration, AE values vary between 0.41 and 0.74 eV in temperatures between 0–40 ºC

(Gillooly et al., 2005), while for photosynthetic processes, the AE predicted is lower, AE ~ 0.32 eV

(Allen et al., 2005). From a thorough compilation of data obtained for a wide range of marine systems

(from polar to subtropical and tropical oceanic regions), Regaudie-de-Gioux and Duarte (2012) found

that overall, the Chl-standardized activation energies for photosynthetic production (GPP) varied

between  0.29–0.32 eV, and for respiratory processes (CR) between 0.65 and 0.66 eV. More recently,

however, Garcia-Corral et al. (2017) reported activation energies much higher for the subtropical and

tropical ocean, with AE values of 0.97 eV for GPP and 1.26 eV for CR.

We found that AE for Chl-*a* standardized GPP and CR rates were at the upper boundary of

values reported for metabolic reactions (0.2 and 1.2 eV) (Gillooly et al., 2001; Dell et al., 2011),

consistent with the findings that warm ocean ecosystems are characterised by elevated activation

energies (Garcia-Corral et al. 2017). The AE for GPP of 1.2 ± 0.17 eV obtained for the Red Sea was



higher than the overall value predicted by the MTE. Surprisingly, however, AE values for CR were

below those for GPP, at $0.73 \pm 0.17$ eV, unlike observed elsewhere in the ocean (Regaudie-de-Gioux

and Duarte 2011, Garcia-Corral et al. 2017). Furthermore, these AE values imply that GPP rates

increased faster (5.1-fold) than CR rates (2.7-fold), from 22–32.5 ºC, the thermal range we measured in

Red Sea waters. These findings differ with the expected double increase of heterotrophic respiration

(regarding photosynthetic processes) with temperature (Harris et al., 2006), but are closer to results

obtained from oligotrophic subtropical and tropical regions (Garcia-Corral et al., 2017).

The apparent contradiction between our findings and the general patterns predicted by the MTE

is, however, not surprising. In their model, Allen et al. (2005) predict the activation energy of

photosynthesis per chloroplast (for temperatures between 0–30 ºC) using the temperature dependence

parameters obtained by Bernacchi et al. (2001) for RuBisCO carboxylation rates in one species (tobacco

leaves). Although the temperature range selected by Allen et al. (2005) comprises the optimum

temperatures of growth rates for a wide range of functional groups of marine primary producers (Chen,

2015; Thomas et al., 2016), the temperature observed in the Red Sea exceeded this range. Due to the

fast generation times of microbes (Collins, 2010), we can expect that photosynthetic planktonic

communities are acclimated or even locally adapted to the thermal conditions they experience. So by

favouring certain photosynthetic or thermal traits, they can enhance their metabolism and growth to the

temperatures they experience, up to their thermal optimum (Galmes et al., 2015; Thomas et al., 2016). It

is therefore likely that the acclimation or local adaptation (in the long term) of photosynthetic traits of

Red Sea plankton optimises the metabolic response at the high-temperatures reached there, resulting in

a steeper response to temperature than predicted by the MTE. Moreover, as the trait responses to



temperature vary among phylogenetic groups (Galmes et al., 2015; Galmés et al., 2016; Thomas et al., 2016), we anticipated a certain degree of discrepancy if we characterise the photosynthetic response (GPP) of planktonic communities forming an ecosystem, by considering only one trait (i.e., RuBisCO carboxylation) of one species.

However, we must consider that the metabolic response of individuals is not only temperature-dependent, and resource supply also plays an essential role (Brown et al., 2004; Allen and Gillooly, 2009); thus their effect on planktonic communities can be intertwined (Raven and Geider, 1988; Marañón et al., 2014). A closer look at our results evidenced that the increased response of planktonic metabolism towards warmer temperatures was also linked to higher autotrophic biomass, and mostly

confined to the southern half of the Red Sea (below 21–22 ºN). This region receives the direct inflow of the enriched Intermediate Water from the Gulf of Aden during the winter monsoon (Raitsos et al., 2015; Wafar et al., 2016). These findings are in line with work demonstrating an increased metabolic response (i.e., mass-specific carbon fixation) of phytoplanktonic communities with temperature when nutrients are not limiting their growth (Marañón et al., 2014; Marañón et al., 2018).

Therefore, unlike the global ocean, where nutrient concentration is inversely correlated with temperature (e.g., Agawin et al. 2000), in the Red Sea nutrient concentration and temperature are positively correlated. This anomaly may explain the steep AE for GPP, as primary producers in the warmer region are being supported by the inflow of the nutrient-enriched waters from the Indian Ocean. The elevated AE for GPP compared to CR in Red Sea plankton is also an anomaly, likely associated

with the lack of allochthonous nutrient supply due to the absence of rivers and vegetation in the arid watershed of the Red Sea. The warm oligotrophic ocean is characterised by plankton communities that




are in metabolic balance or net metabolic imbalanced (Duarte and Agusti 2008, Duarte et al. 2013). In

contrast, the warm Southern Red Sea tends to support autotrophic metabolism, sustained by the input of

nutrient-rich waters while low allochthonous carbon inputs may constrain CR. As a result, NCP tends to

increase, rather than decrease with increasing temperature (Regaudie-de-Gioux and Duarte 2011,

Garcia-Corral et al. 2017). Hence, the patterns in plankton metabolism in the oligotrophic and warm

Red Sea deviate from those characterising the subtropical and tropical gyres of the open ocean. This

anomaly provides an opportunity to explore the mechanistic basis for the global patterns in plankton

metabolism with temperature, which would otherwise remain obscured with the underlying prevalent

negative relationship with nutrient concentrations.

Despite the lack of significant relationship between nutrient availability with metabolic rates, the

close relationship found between planktonic metabolic rates with autotrophic biomass (Chl-*a*), and the

significant relationship between Chl-*a* concentration and nutrient availability, suggest that variables

regulating phytoplanktonic metabolism are also defining the metabolic response of the rest of the

planktonic organisms in the region. The overall low nutrient concentration in the mixed layer along the

basin (except a few stations in the south during autumn), and the relatively high Chl-*a* concentration

during autumn and winter in the Southern Red Sea suggest fast turnover rates of the nutrient pools, a

common feature in oligotrophic environments (Capblancq, 1990). Hence, Red Sea plankton

communities are likely to be efficient using nutrients as they seemed to be rapidly consumed, with an

associated response of autotrophic and heterotrophic communities (i.e., increase in GPP and CR rates).





## 5. Conclusions

Our results show that plankton metabolism in the Red Sea presents a remarkably different

pattern compared to other warm and oligotrophic marine systems (e.g., the subtropical and tropical

gyres). In this region, autotrophic plankton communities prevailed and are supported by relatively high

GPP rates; above the threshold separating heterotrophic low-productivity communities from autotrophic

ones. Metabolically-balanced or heterotropic plankton communities dominated in the Northern Red Sea,

whereas autotrophic communities, supported by nutrient inputs from the Gulf of Aden, were

predominant in the south. Elevated temperatures contributed to an enhanced metabolic activity of

planktonic organisms due to the increase in kinetic energy (favouring enzymatic reactions) with

temperature. Plankton communities in the Red Sea, however, displayed AEs for GPP that were higher

than those for CR, resulting in a positive relationship between NCP and temperature. Those findings

represent anomalies in the relationship between metabolic rates and temperature compared to the warm,

oligotrophic open ocean. These anomalies are likely related to the higher nutrient supply from nutrient-

rich Indian Ocean waters in the warm Southern Red Sea, suggesting that GPP can respond strongly to

the temperature in the warm ocean when supported by high nutrient inputs, relative to those in the

subtropical gyres.

### Author Contributions

DCL-S, CMD, and SA designed the study; KR and PCdA obtained the data and provide technical

support; DCL-S analysed the data; DCL-S wrote the article with a substantial contribution of CMD, and

SA; all authors discussed the results and commented on the manuscript.





**Acknowledgements**

The research reported in this publication was supported by funding from King Abdullah University of

Science and Technology (KAUST), under award number BAS/1/1071-01-10 assigned to CMD,

BAS/1/1072-01-01 assigned to SA, and FCC/1/1973-21-01 assigned to the Red Sea Research Center.



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

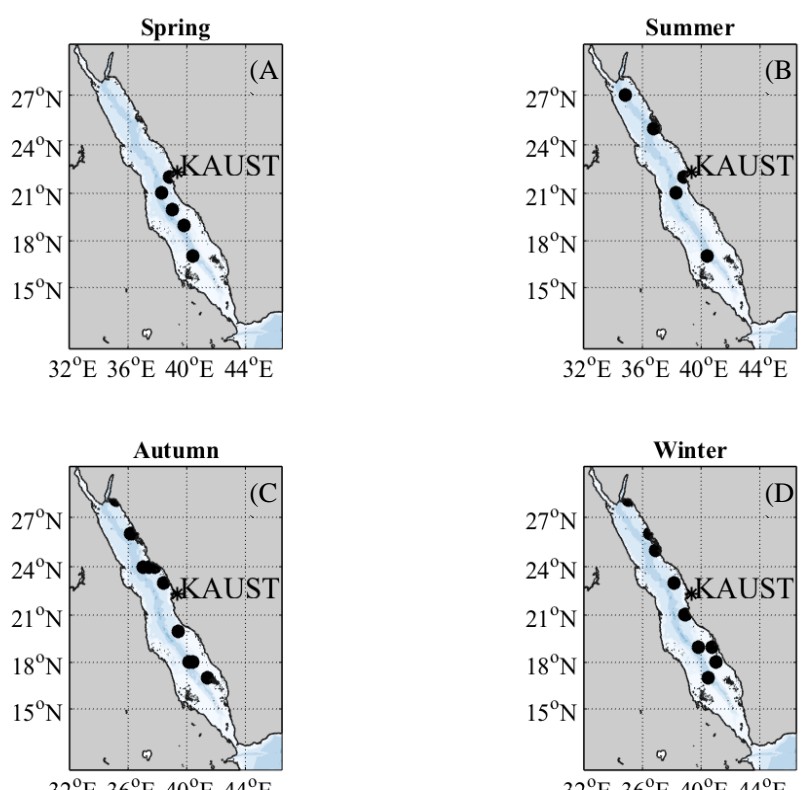

Figure 1: Stations sampled along the Red Sea during (A) spring (2018), (B) summer 2018, (C) autumn (2016) and (D) winter 2016 and 2017





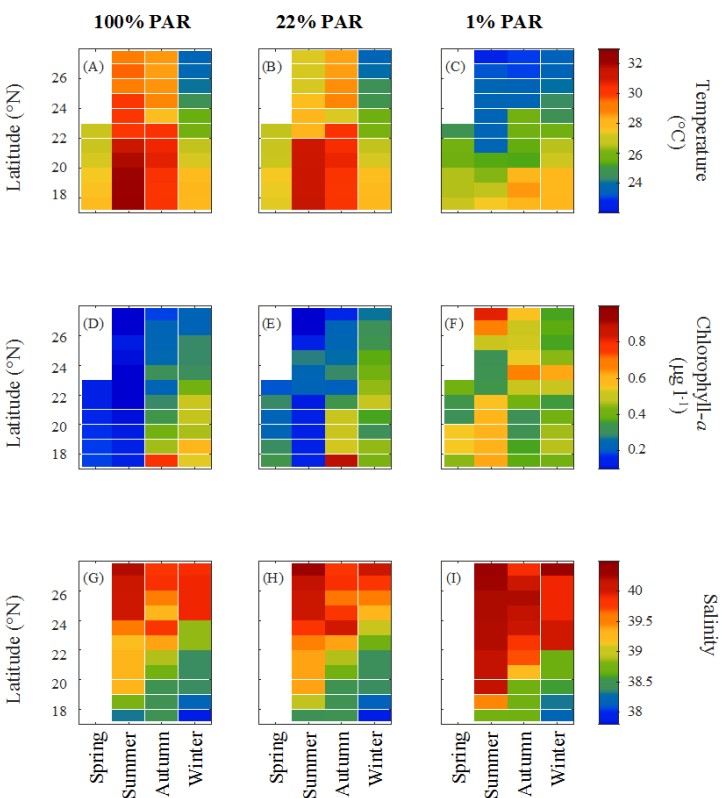

Figure 2: Variability of temperature (ºC) (A–C), chlorophyll-a concentration (D–E), and salinity (G–F) measured during the spring (2018), summer (2017), autumn (2016) and winter (2016 and 2017) cruises along the Red Sea at three different optical depths; 100%, 22 % and 1 % of the photosynthetically active radiation, PAR.





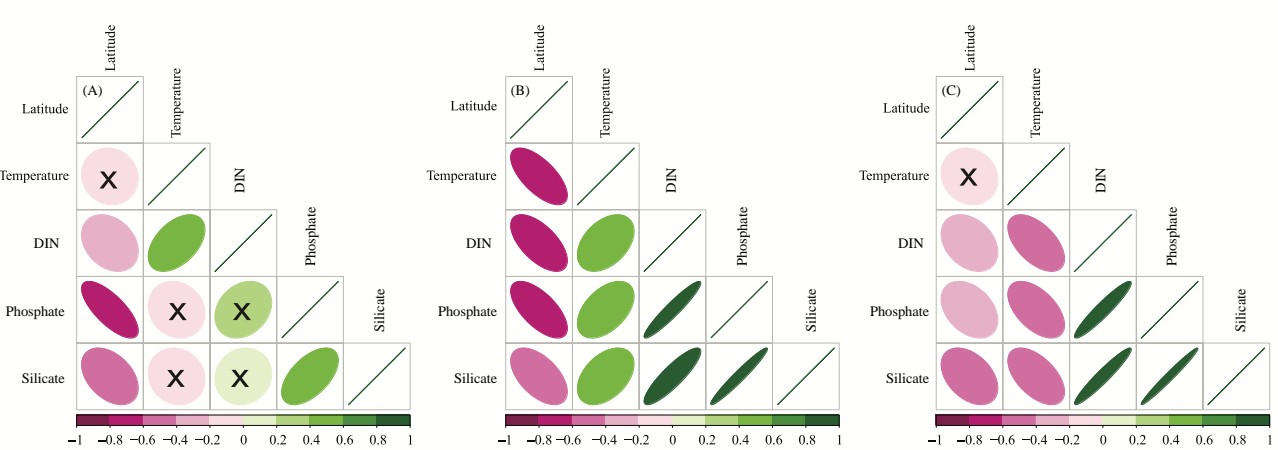

Figure 3: Pearson correlation between the latitude (ºN), temperature (ºC), and the concentration (μM) of dissolved inorganic nitrogen (DIN, NO3 +NO2), phosphate and silicate at (A) within the first optical depth (37 % PAR), (B) at the base of the euphotic layer (1–0.1% PAR), and (C) below the euphotic layer (< 0.1 % PAR). Non-significant correlations are marked with an x.

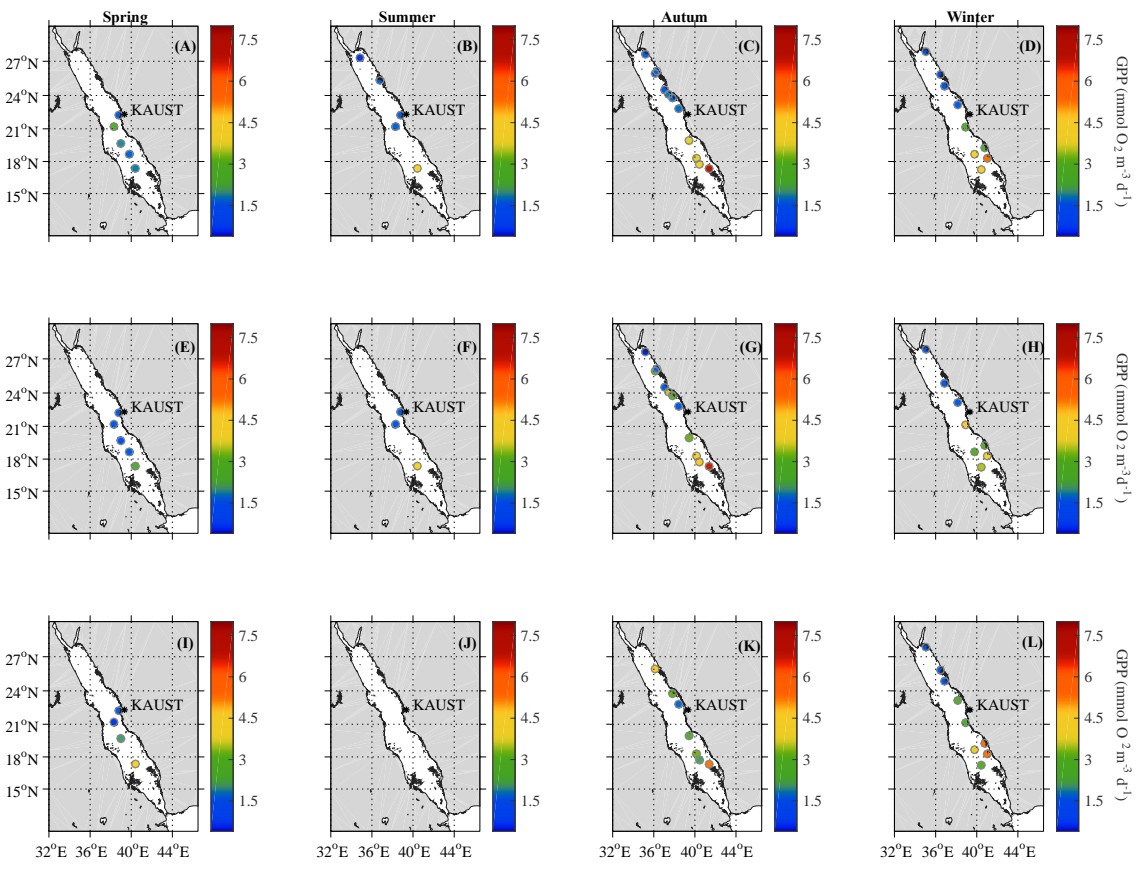

Figure 4: Gross primary production (GPP) measured during the spring (2018), summer (2017), autumn (2017) and winter (2016 and 2017) along the Red Sea at three different optical depths. (A-D) 100 % of the photosynthetically active radiation, PAR. (E-H) 60–8 % of PAR, and (I–L) 1 % of PAR.





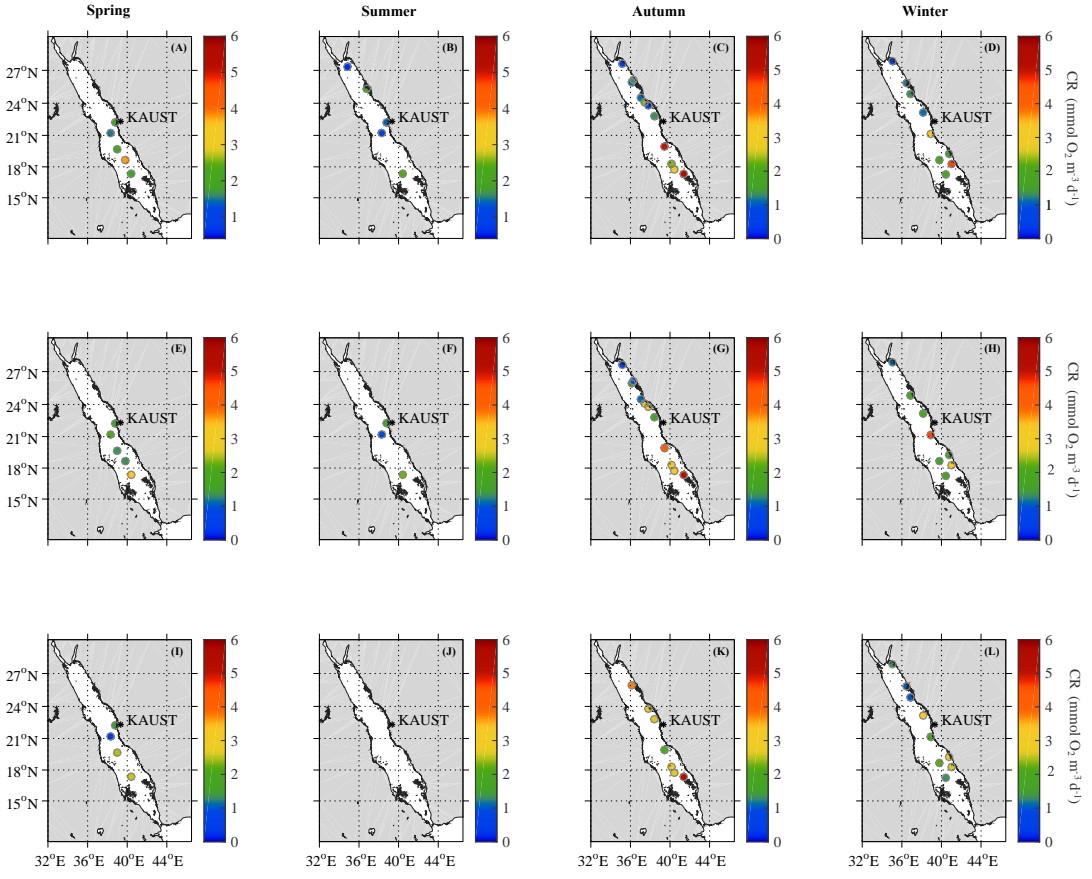

Figure 5: Community respiration rates (CR) measured during the spring (2018), summer (2017), autumn (2017) and winter (2016 and 2017) along the Red Sea at three different optical depths. (A-D) 100 % of the photosynthetically active radiation, PAR. (E-H) 60–8 % of PAR, and (I–L) 1 % of PAR.





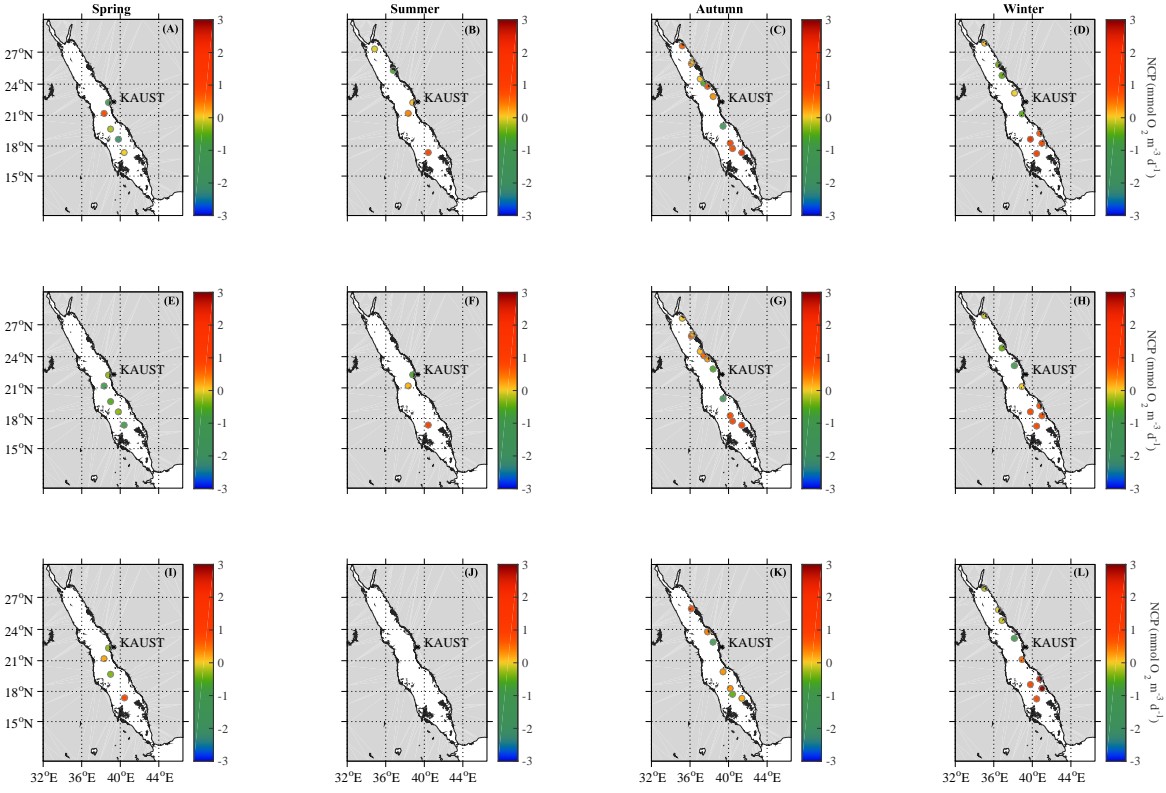

Figure 6: Net community production (NCP) measured during the spring (2018), summer (2017), autumn (2017) and winter (2016 and 2017) along the Red Sea at three different optical depths. (A-D) 100 % of the photosynthetically active radiation, PAR. (E-H) 60–8 % of PAR, and (I–L) 1 % of PAR.



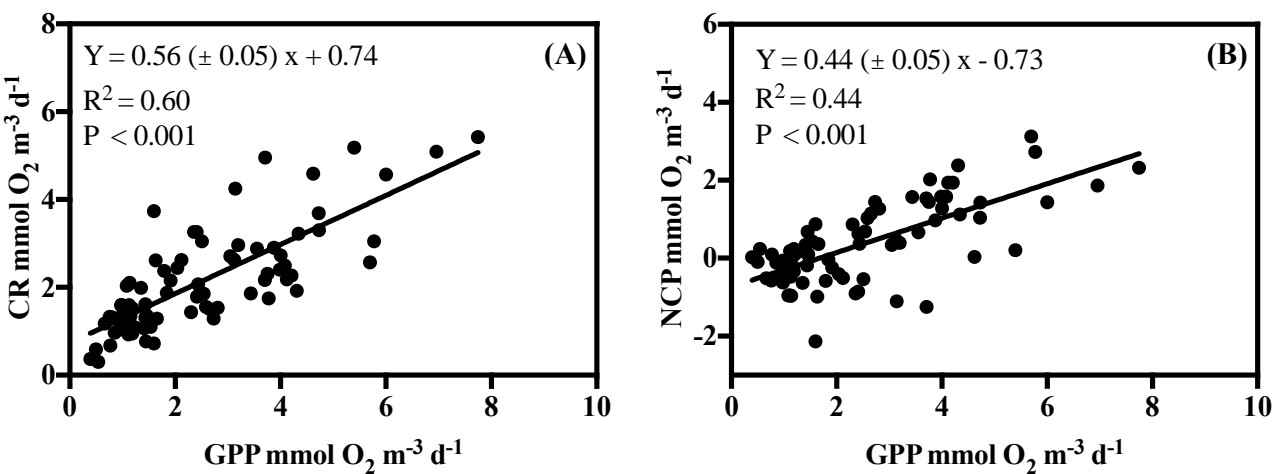

Figure 7: Relationship between (A) planktonic community respiration and (B) net community production (NCP) with gross primary production (GPP) rates measured along the Red Sea. The ordinary least square regression parameters (slope and intercept) and the statistical significance of the correlation (P value) are indicated. The regression models obtained by reduced major axis (RMA) analysis are CR = 0.71 (GPP) + 0.35, and NCP = 0.62 (GPP) -1.19.





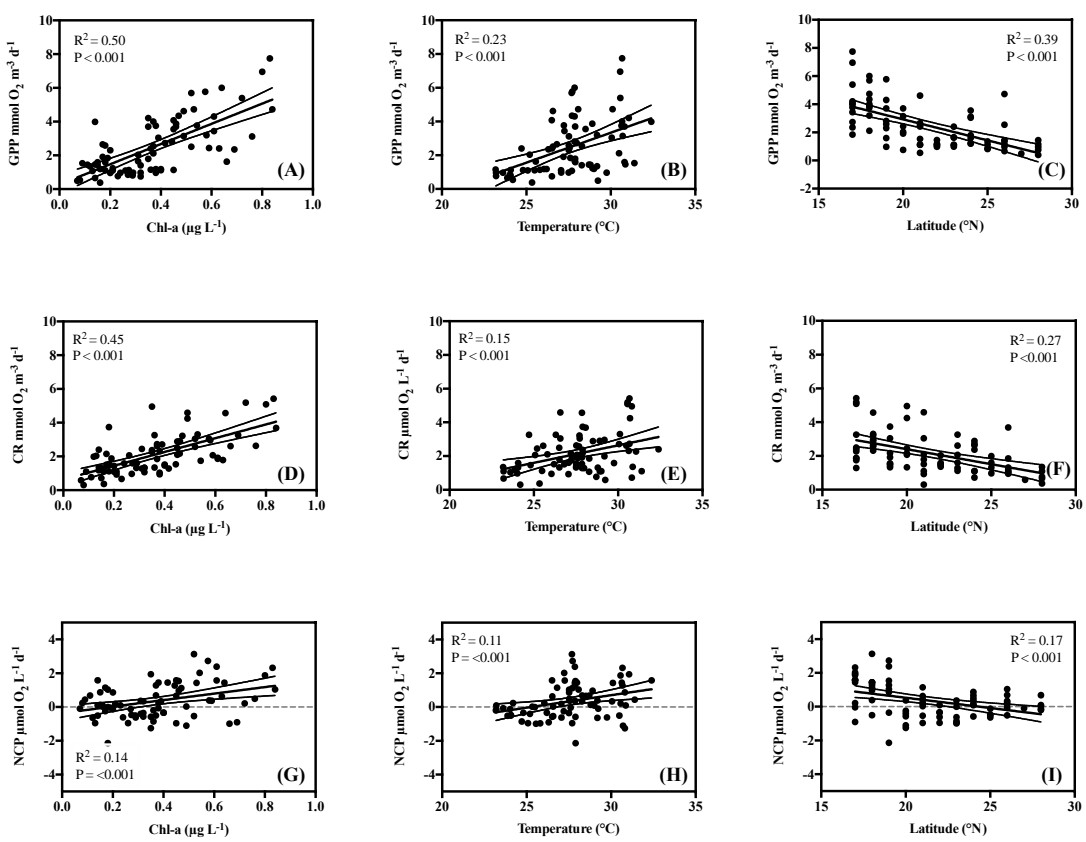

Figure 8: Relationship of gross primary production (GPP), planktonic community respiration (CR) and net community production rates (NCP) with Chlorophyll-*a* concentration (A, D, G), temperature (B, E, H) and latitude (C, F, I)





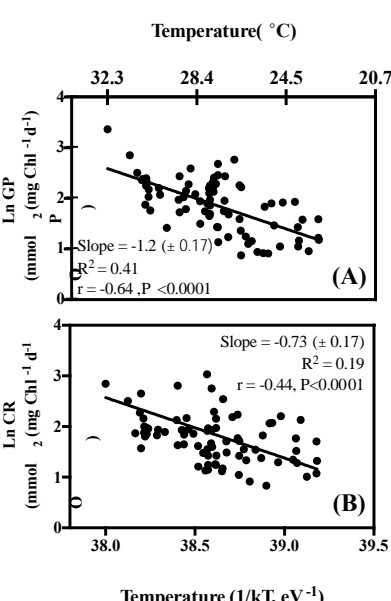

Figure 9: Relationship between the natural logarithm of chlorophyll-*a* normalised gross primary production (A), and planktonic community respiration (B) with temperature as a function of 1/kT (lower axis). The corresponding temperatures in degree Celsius are presented in the upper axis. The solid line is the fitted OLS linear regression. In the function 1/kT, k is the Boltzmanss's constant (8.2 x $10^{-5}$ eV $K^{-1}$), and T is the absolute temperature (°K). The slopes obtained by RMA analysis are -1.8 for (A) and -1.1 for (B).



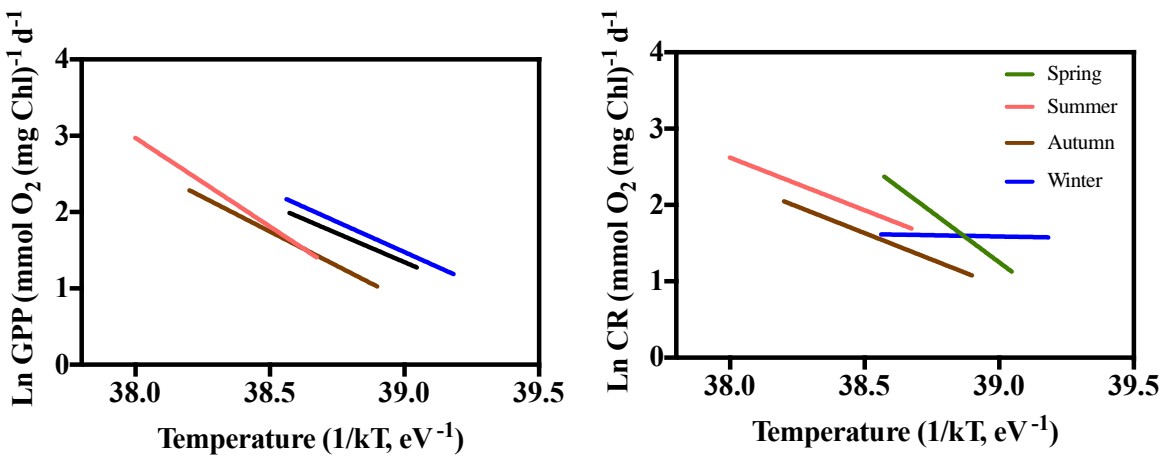

Figure 10: Seasonal relationship between the natural logarithm gross primary production (A) and planktonic respiration rates (B) normalised by chlorophyll-*a* concentration with temperature as a function of 1/kT. Each line represents the fitted OLS linear regression.



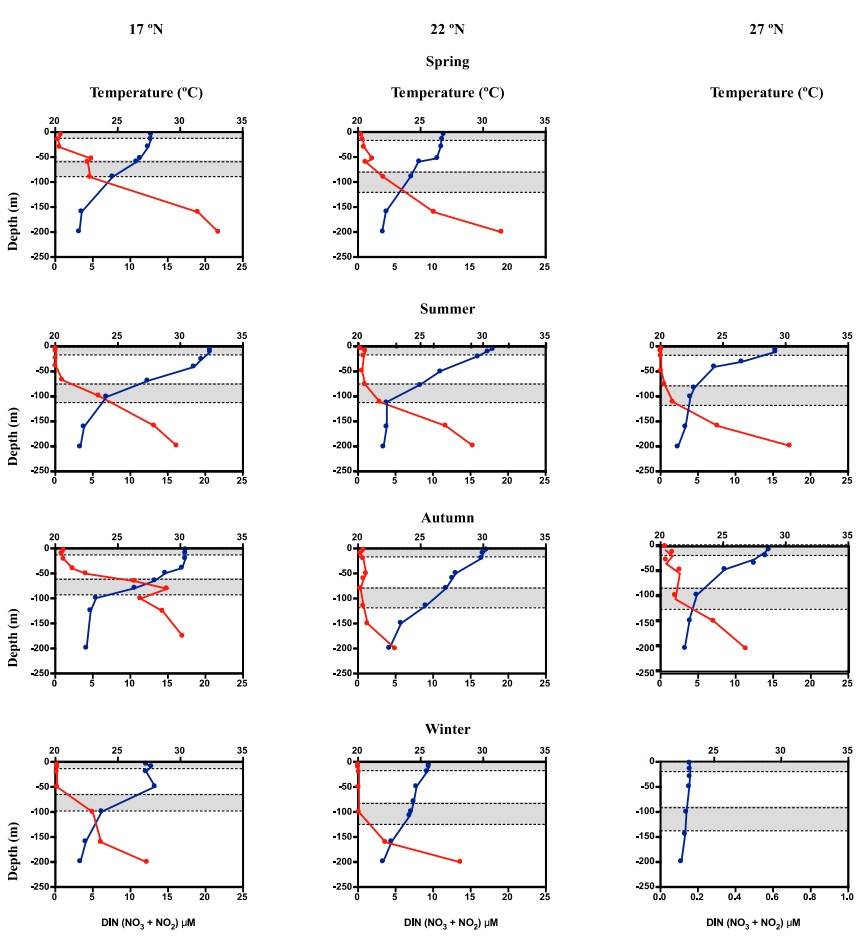

Figure A1: Vertical profiles of temperature (ºC) (blue line) and dissolved inorganic nitrogen concentration (DIN, μM) (red line) measured during March 2018 (spring), August 2017 (summer), November 2016 (autumn) and January 2017 (winter) along the Red Sea. Depicted in grey are the first optical depth (up to 37 % of PAR) and the limit of the euphotic layer (defined between 1–0.1 % of PAR)





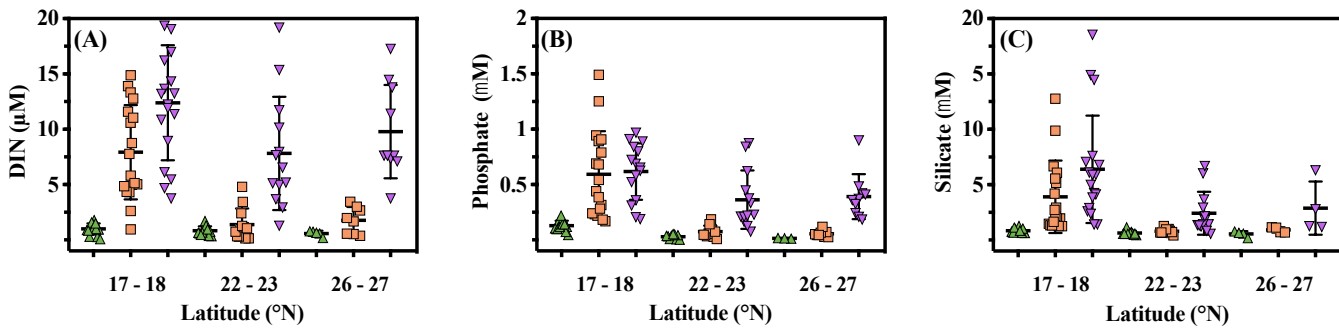

Figure A2: Overall concentration of (A) dissolved inorganic nitrogen (DIN, $NO_2 + NO_3$), (B) phosphate and (C) silicate measured between 17–18 °N, 22–23 °N and 26–27 °N in the Red Sea at different optical depths. Green triangles correspond to measurements within the first optical depth ( up to 37 % PAR), orange squares correspond to measurements at the base of the euphotic zone (1–0.1 % PAR ), purple triangles correspond to measurements below the photic layer (< 0.1 % PAR) at 200 m.





Table 1: Variability of dissolved inorganic nitrogen (DIN, NO2 + NO3) and phosphate concentration measured at different optical depths along the Red Sea during March 2018 (spring), August 2017 (summer), October and November 2017 (autumn) and January 2017 (winter).  PAR is the photosynthetically active radiation. - when data were below the detection limit (details in methods).

| Season | N | DIN (µM) | | | | Phosphate (µM) | | | | Silicate (µM) | | | |
|---|---|---|---|---|---|---|---|---|---|---|---|---|---|
| | | mean | SE | min | max | mean | SE | min | max | mean | SE | min | max |
| 37 % PAR | | | | | | | | | | | | | |
| Spring | 8 | 0.71 | 0.17 | 0.10 | 1.50 | 0.11 | 0.06 | 0.05 | 0.22 | 0.85 | 0.07 | 0.57 | 1.13 |
| Summer | 8 | 0.61 | 0.11 | 0.18 | 1.02 | 0.02 | 0.02 | 0.01 | 0.04 | 0.80 | 0.03 | 0.69 | 0.90 |
| Autumn | 25 | 0.96 | 0.08 | 0.36 | 1.79 | 0.06 | 0.05 | - | 0.18 | 0.64 | 0.03 | 0.44 | 1.14 |
| Winter | 5 | 0.40 | 0.15 | 0.10 | 0.98 | 0.09 | 0.07 | 0.01 | 0.17 | 0.84 | 0.14 | 0.52 | 1.26 |
| 15–5 % PAR | | | | | | | | | | | | | |
| Spring | 4 | 0.89 | 0.40 | 0.02 | 1.94 | 0.08 | 0.02 | 0.06 | 0.13 | 0.85 | 0.07 | 0.69 | 1.05 |
| Summer | 4 | 0.12 | 0.04 | 0.06 | 0.23 | 0.03 | 0.01 | - | 0.06 | 0.88 | 0.02 | 0.84 | 0.93 |
| Autumn | 18 | 1.31 | 0.18 | 0.60 | 3.24 | 0.05 | 0.01 | 0.01 | 0.22 | 0.69 | 0.04 | 0.46 | 1.26 |
| Winter | 3 | 0.38 | 0.19 | 0.12 | 0.75 | 0.08 | 0.03 | 0.02 | 0.14 | 0.76 | 0.06 | 0.68 | 0.88 |
| 1–0.1 % PAR | | | | | | | | | | | | | |
| Spring | 9 | 3.65 | 0.56 | 1.06 | 5.79 | 0.17 | 0.03 | 0.05 | 0.28 | 1.19 | 0.10 | 0.69 | 1.65 |
| Summer | 9 | 1.53 | 0.59 | 0.36 | 5.82 | 0.14 | 0.04 | 0.03 | 0.44 | 1.22 | 0.12 | 0.85 | 1.97 |
| Autumn | 26 | 5.47 | 0.96 | 0.30 | 14.88 | 0.40 | 0.09 | 0.01 | 1.49 | 2.76 | 0.62 | 0.43 | 12.76 |
| Winter | 6 | 1.41 | 0.82 | 0.06 | 5.00 | 0.09 | 0.05 | 0.01 | 0.31 | 0.94 | 0.23 | 0.51 | 1.94 |





Table 2: Gross primary production (GPP), community respiration (CR), net community production, production to respiration ratio (P:R) and the percentage of heterotrophy (NCP < 0) measured at different optical depths along the Red Sea during March 2018 (spring), August 2017 (summer), October and November 2017 (autumn), and February 2016 and January 2017 (winter). PAR is the photosynthetically active radiation.

| % PAR | | GPP | | | | CR | | | | NCP | | | | P: R | | | | NCP < 0 |
|---|---|---|---|---|---|---|---|---|---|---|---|---|---|---|---|---|---|---|
| 100–37 | n | mean | SE | | Rank | mean | SE | | Rank | mean | SE | | Rank | mean | SE | | Rank | % |
| Spring | 5.00 | 1.82 | 0.26 | 1.08 | 2.65 | 2.26 | 0.38 | 1.51 | 3.73 | -0.45 | 0.54 | -2.14 | 1.14 | 0.92 | 0.23 | 0.43 | 1.76 | 80 |
| Summer | 5.00 | 1.77 | 0.59 | 0.50 | 3.98 | 1.49 | 0.32 | 0.59 | 2.41 | 0.28 | 0.37 | -0.64 | 1.58 | 1.13 | 0.18 | 0.68 | 1.66 | 40 |
| Autumn | 11.00 | 2.69 | 0.60 | 1.19 | 7.75 | 2.20 | 0.49 | 0.72 | 5.42 | 0.49 | 0.29 | -1.25 | 2.32 | 1.34 | 0.13 | 0.75 | 2.21 | 18 |
| Winter | 13.00 | 2.67 | 0.49 | 0.67 | 6.01 | 2.11 | 0.37 | 0.67 | 4.59 | 0.56 | 0.27 | -0.54 | 2.38 | 1.26 | 0.15 | 0.57 | 2.24 | 38 |
| Overall | 34.00 | 2.42 | 0.29 | 0.50 | 7.75 | 2.07 | 0.22 | 0.59 | 5.42 | 0.35 | 0.17 | -2.14 | 2.38 | 1.22 | 0.08 | 0.43 | 2.24 | 38 |
| **36–6** | | | | | | | | | | | | | | | | | | |
| Spring | 2.00 | 1.08 | 0.10 | 0.98 | 1.18 | 1.43 | 0.10 | 1.33 | 1.52 | -0.35 | 0.01 | -0.35 | 1.14 | 1.09 | 0.33 | 0.73 | 1.76 | 67 |
| Summer | 3.00 | 2.10 | 1.06 | 0.97 | 4.21 | 1.60 | 0.39 | 0.93 | 2.27 | 0.50 | 0.75 | -0.62 | 1.94 | 1.22 | 0.36 | 0.61 | 1.85 | 33 |
| Autumn | 11.00 | 2.89 | 0.58 | 0.39 | 6.96 | 2.44 | 0.43 | 0.37 | 5.09 | 0.45 | 0.26 | -1.10 | 1.87 | 1.16 | 0.08 | 0.61 | 1.47 | 18 |
| Winter | 8.00 | 3.21 | 0.64 | 1.11 | 5.77 | 2.26 | 0.23 | 1.35 | 3.26 | 0.95 | 0.55 | -0.95 | 3.13 | 1.33 | 0.14 | 0.54 | 2.22 | 36 |
| Overall | 24.00 | 2.75 | 0.37 | 0.39 | 6.96 | 2.19 | 0.22 | 0.37 | 5.09 | 0.56 | 0.24 | -1.10 | 3.13 | 1.24 | 0.08 | 0.54 | 2.22 | 32 |
| **5–1** | | | | | | | | | | | | | | | | | | |
| Spring | 7.00 | 1.72 | 0.47 | 0.54 | 4.08 | 1.91 | 0.37 | 0.31 | 3.26 | -0.20 | 0.33 | -0.97 | 1.58 | 0.98 | 0.19 | 0.54 | 1.77 | 71 |
| Autumn | 7.00 | 3.23 | 0.52 | 1.63 | 5.40 | 3.11 | 0.39 | 2.07 | 5.19 | 0.14 | 0.26 | -0.99 | 1.04 | 1.03 | 0.09 | 0.62 | 1.28 | 29 |
| Winter | 4.00 | 1.42 | 0.44 | 0.85 | 2.73 | 1.23 | 0.14 | 0.97 | 1.60 | 0.20 | 0.43 | -0.47 | 1.44 | 1.16 | 0.32 | 0.70 | 2.12 | 75 |
| Overall | 18.00 | 2.24 | 0.33 | 0.54 | 5.40 | 2.23 | 0.27 | 0.31 | 5.19 | 0.02 | 0.18 | -0.99 | 1.58 | 1.04 | 0.10 | 0.54 | 2.12 | 56 |



Table 3. Correlation and probability matrix of environmental variables and metabolic rates. DIN refers to the dissolved inorganic nitrogen ($NO_3$ + $NO_2$) concentration (mM). Chl-*a* is the the concentration of Chlorophyll-*a* (mg l$^{-1}$), GPP is the gross primary production, CR is the community respiration and NCP is the net community production (all in mmol O$_2$ m$^{-3}$ d$^{-1}$). Temperature is in (ºC).

| | GPP | CR | NCP | Season | Latitude | DIN | Depth | Temperature | Chl-a |
|---|---|---|---|---|---|---|---|---|---|
| GPP | 1.00 | 0.79 | 0.70 | 0.24 | -0.62 | 0.00 | -0.11 | 0.48 | 0.71 |
| CR | 0.79 | 1.00 | 0.11 | 0.06 | -0.52 | 0.10 | -0.06 | 0.39 | 0.67 |
| NCP | 0.70 | 0.11 | 1.00 | 0.31 | -0.41 | -0.11 | - | 0.32 | 0.37 |
| Season | 0.24 | 0.06 | 0.31 | 1.00 | - | -0.16 | - | -0.16 | 0.25 |
| Latitide | -0.62 | -0.52 | -0.41 | - | 1.00 | -0.01 | 0.03 | -0.44 | -0.40 |
| DIN | 0.00 | 0.10 | -0.11 | -0.16 | -0.01 | 1.00 | 0.35 | -0.07 | 0.26 |
| Depth | -0.11 | -0.06 | -0.11 | - | - | 0.35 | 1.00 | -0.39 | 0.30 |
| Temperature | 0.48 | 0.39 | 0.33 | -0.16 | -0.44 | -0.07 | -0.39 | 1.00 | 0.10 |
| Chl-a | 0.71 | 0.67 | 0.37 | 0.25 | -0.40 | 0.26 | 0.30 | 0.10 | 1.00 |
| | GPP | CR | NCP | Season | Latitude | DIN | Depth | Temperature | Chl-a |
| GPP | <0.001 | <0.001 | <0.001 | 0.04 | <0.001 | 0.99 | 0.35 | <0.001 | <0.001 |
| CR | <0.001 | <0.001 | 0.33 | 0.59 | <0.001 | 0.47 | 0.62 | <0.001 | <0.001 |
| NCP | <0.001 | 0.33 | <0.001 | 0.01 | <0.001 | 0.43 | 0.36 | <0.001 | <0.001 |
| Season | 0.04 | 0.59 | 0.01 | <0.001 | - | 0.25 | - | 0.16 | 0.03 |
| Latitide | <0.001 | <0.001 | <0.001 | - | <0.001 | 0.96 | - | <0.001 | <0.001 |
| DIN | 0.99 | 0.47 | 0.43 | 0.25 | 0.96 | <0.001 | 0.01 | 0.62 | 0.05 |
| Depth | 0.35 | 0.62 | 0.36 | - | - | 0.007 | <0.001 | <0.001 | 0.01 |
| Temperature | <0.001 | <0.001 | <0.001 | 0.16 | <0.001 | 0.62 | <0.001 | <0.001 | 0.39 |
| Chl-a | <0.001 | <0.001 | <0.001 | 0.03 | <0.001 | 0.05 | 0.01 | 0.39 | <0.001 |

\* The correlations without an ecological significance are left in blanc (-)