# Peer review of "Rates and drivers of Red Sea plankton community metabolism"

_Biogeosciences, 2018_

## Referee Comment (RC1) · Anonymous Referee #1 · 21 Dec 2018

General comments: The authors describe a dataset of environmental variables related to the metabolism of planktonic communities along a depth and latitudinal gradient in a seasonal resolution in the Red Sea. The authors conclude that gross primary production relates positively to sea surface temperature and nutrient availability. The dataset is extensive, and the research questions (for this part of the Red Sea), to my knowledge, are novel and worthy of publication. The abstract is clear and reads well, but shows a different narrative than the rest of the manuscript. Thus, I suggest for the authors to consider rewriting the manuscript. As mentioned in the author contributions, the manuscript is written by several people and this is noticeable (see specific comments). The abstract mentions the latitudinal gradient but the ms introduces two more variables, i.e. depth and seasonality. While interesting variables, they make the

story confusing at times and harder to disentangle the story the authors want to tell (according to the abstract).

Concerns about the methods used are mentioned in the specific comments and need to be addressed first. Proper description of statistical analyses is lacking.

My recommendation is that the ms needs major revisions, but only if methodological concerns can be addressed adequately. Then, I suggest a complete overhaul of the manuscripts narrative by focusing on 1 or 2 of the 3 major variables (latitude, water depth and seasonality) and stick with these in the entire narrative. Also, there needs to be a clear description of used statistics in the M&M section and figures and tables should be cut back and/or improved. Consistency in the presentation of the results (including the statistics) and the use of abbreviations (as well as changing them) is recommended.

Specific comments: - Title: Says Red Sea but Gulfs are not included. - Abstract: Line 10: Mentioning "Low productive waters" immediately brings down the importance of the story. - Page 2: The first paragraph is loaded with self-referencing while many others are not or less. - Page 2, line 4-5 and 11: Introduce abbreviations once (see technical corrections) and use them consistently throughout the ms. - Page 2: The abbreviations of GPP, CR and NCP are presented with units of daily oxygen produced or used. However, these abbreviations are normally used for daily production and use of carbon. I suggest the authors change the abbreviations for these processes and/or use a conversion factor to present daily carbon production and use. - Page 4, line 7-8: There are plenty of references that describe metabolism in the northern part of the Red Sea (e.g. Rahav et al. 2015 MEPS, Tilstra et al 2018 Frontiers, Levanon-Spanier et al 1979 Deep Sea Res.). - Page 5-6: Silicate is measured, mentioned in the results and in many figures/tables (with significant interactions) but nowhere mentioned in the Discussion. If not important, mention briefly in Discussion. - Page 6-7, line 20 and 1 (resp.): Was NH4 determined? If not, then you have NOx values, not DIN - Page 7, line 10: provide actual depths of PAR measurements. Also, I am confused

about the use of 100%, 60-20 and 8-1 as table 1 and 2 give different ranges. Is the data comparable if different depths of sampling were used? - Page 7: Were samples for metabolic rates filtered? Does the planktonic community include both single and multicellular organisms? Were the optodes adjusted for salinity? - Page 7: A major, potential, flaw in the methods used for metabolic rates is that it appears as though net photosynthesis was measured for 24h. If correct, this includes an approx. 12-hour period of darkness and thus results in data that cannot be used for calculations for gross photosynthesis, i.e. $O_2$ measurements will be severely lower due to dark respiration. net photosynthesis should have been measured only during daylight and respiration rates should have been measured in 2 phases; during the daytime and during nighttime, so approximately 12:12 h as respiration rates can have a diurnal rhythm. So extrapolating these data to daily rates could result in a wrong estimation of gross photosynthesis. Also, how was $O_2$ production data extrapolated to per day? I suggest authors stick to hourly values for oxygen rates. If methods are used correctly, carbon budgets can be calculated using a conversion factor. If net photosynthesis was measured during daylight and respiration for 24 hours, the authors need to state assumptions of the values to their manuscript (potential over- or underestimation of rates). - Page 9, statistics: Need to be expanded with actual models used. - Page 9, line 11: NOx, not DIN. - Page 10, line 17: 56% of heterotrophs suggests dominance of this trophic strategy - Page 11, line 4: What models were used to test this? - Page 11, line 8: Which analysis? - Page 11, line 11: Introduction of this statistical method should be in the appropriate section - Page 11: How were AE values calculated? - Page 11: AE are presented as negatives, are they? Next page the authors mention a positive value. - Page 13, line 14: GPP is said to be low, compared to what? - Page 15, line 18: How was AE standardized to Chl-a? - Page 16, line 2: What is "the ocean"? - Page 16: Opens with "Surprisingly" and a discussion, then the next paragraph mentions a contradiction that is not surprising. What is the contradiction exactly the authors mean? - Page 16, line 6-7: Authors compare results with other references but need to mention actual values. - Figure 3: Thickness of the pink or green seems to say something about

how significant it is but this is said nowhere. In line with this, the diagonal dark green lines seem to signify extreme significance instead of same variable and thus not tested. DIN is NOx. Are variables tested at different depths than metabolic rates of plankton? If so, how can you relate the 2? - Figure 4-6: Lots of white space and hard to see with tiny colored dots anyway. Revise these figures. I suggest to distill from them the most important results you want to show and add the rest to the supplementary section. - Figure 7: could be mentioned with text in the results section. Suggest moving figure to supplements. - Figure 9: Same as Figure 7, B is missing a parenthesis on the y-axis - Figure 10: Same as Figure 7

Technical corrections: - Please use continues line numbers for the manuscript - Page 1, line 8-9: Please rewrite, it reads as if you want to understand their variability and their present and their future but you want to understand their variability in the present and the future - Page 2, line 4-5: Add community - Page 2, line 11: First mention of NCP, introduce abbreviation - Page 3, line 1: "The Red Sea is a semi-enclosed" - Page 3, line 3-5: Consider merging this sentence with the previous one - Page 3, line 9: "throughout the year" - Page 3, line 10: Delete the dot before the references - Page 4, line 12: Add "relatively" to "unproductive waters" - Page 4, line 18: Add "latitudinal gradient" to the sentence - Page 10, line 8-10: I suggest to start the Results section with this sentence. - Page 10, line 16: net autotrophic? - Page 12, line 8: Please stay consistent, use R2 - Page 13, line 9: Heterotrophic suggest no autotrophs, add "net" - Page 15, line 6-9: Please rewrite - Page 16, line 4: Add i.e. or parentheses after 32.5 °C - Page 19, line 6: Heterotrophic - Figure A1: Add axis titles to every part of the figure, having double axes without titles is confusing, especially since the 27 °N axis title (Temperature) is not on any axis. - Table 1: Add Silicate to the table description. Also, it is unclear which header belongs to which environmental variable. Also, I fail to see the benefit of the min and max values. Delete and/or add to supplements. Present data as mean ± SE - Table 2: N does not need decimals. What does "rank" mean? %PAR differs from Table 1. - Table 3: Upper part are, what seems to be, Pearson rank coefficients, not the units given in the description. The lower part seems to be p-values,

mention this in the description. A hyphen is not the same as a blanc.

---

## Referee Comment (RC2) · Anonymous Referee #2 · 14 Jan 2019

The authors quantified plankton metabolic rates along the Red Sea. They have shown that Chla and plankton community metabolism (GPP and CR) increase with temperature. Contrary to previous results, they have observed a higher Activation Energy for GPP than for CR showing a positive relationship between NCP and Temperature. These results have been explained by the authors as a consequence of the high nutrient availability in warmer waters and the lack of external organic carbon sources to sustain a heterotrophic metabolism constraining the CR.

The dataset are very interesting and merit been published, however, the way how the results have been presented, the lack of statistical analyses and the methodology proposed are not the most suitable to achieve the main goal proposed in the manuscript. Therefore, I consider the ms still needs major revision in order to be published and

providing the authors follow the reviewers recommendations.

First, according to the title and the abstract the authors consider as drivers of the plankton community metabolism in the Red Sea, the Chla and temperature. However, other important parameters such as, temporal and spatial variability, salinity and nutrients seem to govern the plankton community metabolism within this particular ecosystem and are not included in the abstract. Therefore, this lack of agreement between the ms, the conclusion and the abstract is confusing. In my opinion, there is a large flaw in the experimental design proposed and it is difficult to resolve. All samples included the deepest ones have been incubated on deck with surface water. During some of the surveys, there is an important thermal variability. The authors have attempted to mitigate the issue by including just those samples above the thermocline. However, Material and Methods mention that changes in temperature and PAR in the incubation tanks were recorded, with HOBO data loggers. Therefore, those data should be shown, in a table in order to select objectively the samples for the analyses. Hence, eliminating those samples that register thermal differences above 2°C with the in situ temperatures. In addition, samples adapted to cool temperatures such as those at the bottom will respond more drastically to artificial increments of temperature than surface ones (for example. Apple et al. 2006. AME. 43: 243–254) resulting in erroneous conclusions. Therefore, Figure A1 is important and should be included in the Ms.

Other figures such as 4-6 do not show crucial information in the current format.

Figure 3 and Table 3 to me are redundant.

The paragraph 10-15 page 6 the authors should indicate if samples were collected before sunrise (to avoid any light on the samples) and if the incubation started at the sunrise to estimate the full light period. The authors say, the samples were collected between 7 to 9 and to me this sounds very late to incubate and obtain the full light period not precisely.

In The net community metabolism..... page 7, NCP should be estimated during the light

period (NCP 6 to 12 hours). The authors should show, the variation coefficient of the pool data and also the original CR, NCP and GPP data including their SE. Because, in these oligrotrophic areas the metabolic rates are very low and can be difficult to detect them. Therefore, the methodology needs to be very precise in the processes of filling, incubating and fixing the bottles.

The paragraph 20 in page 8. It should be indicated the Arrhenius plots the authors mention.

The paragraph 10 in page 10. It should be transfered from the Results to the Discussion. And also, the first paragraph of the 3.2 Variability of plankton …. Is already mentioned in M and M.

The paragraph 10 in page 13. There are lots of references within oligotrophic areas very interesting and different to the authors ones that the authors should also include in the MS.

Figure 1. The name of the KAUST is excessive. I would include a unique bigger map with different colours or shapes to show the stations at each survey or season.

Figure 2. I consider in this figure is difficult to detect the thermocline and the vertical profiles of Chla and salinity. I consider that nutrient profiles should also be included.

Figure 8, 9 and 10. To test one of the main conclusions, if AE is higher for GPP than for CR, authors should test statistically if the slopes are different. I would test also the slopes for the figures 9 and 10 explaining the consequences of the statistical differences in the cases observed. In the figure 9, the RMA analyses have been included but it is not necessary in this case because temperature is not a rate. In addition, the authors have not explained when the RMA or OLS should be used in M and M.

---

## Author Comment (AC1) · 25 Feb 2019

Reviewer 1 General comments: The authors describe a dataset of environmental variables related to the metabolism of planktonic communities along a depth and latitudinal gradient in a seasonal resolution in the Red Sea. The authors conclude that gross primary production relates positively to sea surface temperature and nutrient availability. The dataset is extensive, and the research questions (for this part of the Red Sea), to my knowledge, are novel and worthy of publication. The abstract is clear and reads well, but shows a different narrative than the rest of the manuscript. Thus, I suggest for the authors to consider rewriting the manuscript. As mentioned in the author contributions, the manuscript is written by several people and this is noticeable (see specific comments). The abstract mentions the latitudinal gradient but the ms introduces two

more variables, i.e. depth and seasonality. While interesting variables, they make the story confusing at times and harder to disentangle the story the authors want to tell (according to the abstract). Concerns about the methods used are mentioned in the specific comments and need to be addressed first. Proper description of statistical analyses is lacking. My recommendation is that the ms needs major revisions, but only if methodological concerns can be addressed adequately. Then, I suggest a complete overhaul of the manuscripts narrative by focusing on 1 or 2 of the 3 major variables (latitude, water depth and seasonality) and stick with these in the entire narrative. Also, there needs to be a clear description of used statistics in the M&M section and figures and tables should be cut back and/or improved. Consistency in the presentation of the results (including the statistics) and the use of abbreviations (as well as changing them) is recommended

Reply to General Comments:

We sincerely appreciate the thorough revision, comments and the devoted time to review our manuscript in such a rigorous manner, as the help us to greatly improve our manuscript. We have addressed all the reviewer's comments, and significant changes were done to the figures and tables so that they all consistently provide information from the depths we measured metabolic rates. Also, we have included a detailed description of the statistical analyses. The changes can be tracked in the new version of the manuscript in Figures 2, 3, and Tables 1, 2, and in the methods section 2.4. We summarised the information shown in table 3 in a new figure and moved Figures 4–6 as supplementary information.

Regarding the main concerns about the methodology used to measure planktonic metabolic rates, we think there is a misunderstanding. The reviewer indicated (RC10) that a major potential flaw in our study is the method we used to measure net photosynthesis. We agree with the reviewer that to measure net photosynthesis (i.e., the net organic carbon production in the light, Pn = gross photosynthesis -respiratory losses in the light) (Falkowski and Raven 2007), a 24-hour incubation can carry potential bias.

However, in our study, we are not measuring nor reporting net photosynthetic rates.

We stated in our introduction and methodology that one of our goals was to quantify the metabolic balance of the planktonic community and from there determine the net community production (NCP) and not Pn. NCP is the organic matter remaining after consumption of the GPP through respiration by plants (autotrophs), microbes (either autotrophs or heterotrophs), and animals (heterotrophs) (Ducklow and Doney 2013), and can be derived from the equation GPP = NCP + CR. This equation describes the balance between the photosynthetic process and respiration (Ducklow and Doney 2013). We also would like to highlight that the methodology used to quantify planktonic metabolism is based on the dark and light method in combination with the Winkler titration method (as mentioned in our reply AC5) and not optodes (as the reviewer stated), and that the procedure we followed was explained in detail in the methods section 2.3. The selected methodology based on oxygen measurements is primarily used since the earliest studies of H. H. Gran, and there is a vast amount of literature available where planktonic metabolism is quantified by this method, on a 24-h period, and the results are reported in oxygen units, as it is what is measured.

Reply to specific comments:

Reviewer Comment (RC) Authors Reply (AC)

RC1.- Title: Says Red Sea but Gulfs are not included.

AC1: We agree with the reviewer that our study does not include data from the Gulf of Aqaba nor it does include the western half of the Red Sea. However, this detail is carefully described in the methods section and Fig.1, so the reader will not be misled. Our study focusses on testing the changes across the latitudinal gradient in the Red Sea, where the samples extend between 17–27 °N (while the Gulf of Aqaba, for instance, extends along 28–29 °N). Browsing through the 2018 papers published in Biogeosciences that have one sea or ocean named without any additional qualifications in the title, we can find studies like the one of Sargeant et al., 2018. Their work

was entitled Basin-scale variability of microbial methanol uptake in the Atlantic Ocean. Their work covered a transect from 50° N to 30° S, and therefore missed a large proportion of the Atlantic Ocean, without this generating confusion as to what the results show. We, therefore, prefer to leave the title as it is and provide the details on the sampled areas in the manuscript.

RC2.- Abstract: Line 10: Mentioning "Low productive waters" immediately brings down the importance of the story.

AC2: We modified the text accordingly

RC3.- Page 2: The first paragraph is loaded with self-referencing while many others are not or less.

AC3: There is no intent to load the paper unnecessarily with self-references, but it turns out that the co-authors of this paper have published a large number of relevant articles on the topic. We have now, however, added additional works, by other researchers, to the 1st paragraph.

RC4.- Page 2, line 4-5 and 11: Introduce abbreviations once (see technical corrections) and use them consistently throughout the ms.

AC4: Thank you for your comment, we modified the text accordingly

RC5.- Page 2: The abbreviations of GPP, CR and NCP are presented with units of daily oxygen produced or used. However, these abbreviations are normally used for daily production and use of carbon. I suggest the authors change the abbreviations for these processes and/or use a conversion factor to present daily carbon production and use.

AC5: This and other comments from the reviewer, led us to believe that there is a misunderstanding as they seem to suggest that our work was focus on primary production, which is performed by the photosynthetic components of the plankton community during the daytime and that has gross and net components (as phytoplankton excrete and

respire carbon). However, our paper focuses on the entire plankton community, both photosynthetic and heterotrophic (e.g. bacteria), where the net community production (NCP) represents the organic matter remaining after consumption of the GPP through respiration by plants (autotrophs), microbes (either autotrophs or heterotrophs), and animals (heterotrophs) (Ducklow and Doney 2013). Hence, it is the balance of the photosynthetic processes (GPP) and respiratory activity of the entire plankton community (not just phytoplankton) measured during 24 h periods. Studies that focus on the photosynthetic component of the plankton community (e.g. Net Primary Production, NPP) report values for the daylight period only. Whereas studies, such as this, report (24 h) rates. For instance, published synthesis of community metabolism rates report values per day (24 h), e.g., Robinson and Williams 2005, Regaudie-de-Gioux 2012, 2013).

Reports of daily GPP, NCP and CR rates in oxygen units has remained the case since the Haaken H. Gran, used the light and dark bottle method in combination with the Winkler method to report community metabolic rates (Gaarder and Gran 1927; Gran 1927). The reviewer can find other examples in McIntire et al. 1964; Williams et al. 1979; Williams et al. 1983; Owens and Crumptom 1995; Robinson and Williams 1999; López-Urrutia et al. 2006; Stanley et al. 2010; García-Martín et al. 2017. The use of 24 h to report rates is indeed not just a tradition but is justified as the metabolic budget need be resolved over 24 h to be completed, for photosynthesis this does not proceed at night, but respiration, which is necessary to define net community production, occurs at night as well. Moreover, all of the above papers report rates based on oxygen units, since this is the property measured and converting these to C requires some assumptions, such as a theoretical PQ value. However, to allow comparisons with another component of carbon cycling, we have provided an estimate, as an indication, of what the mean GPP reported here represents in terms of carbon production, assuming a PQ =1 (please refer to this between lines 202-204 in the new version of the manuscript)

RC6.- Page 4, line 7-8: There are plenty of references that describe metabolism in

the northern part of the Red Sea (e.g. Rahav et al. 2015 MEPS, Tilstra et al 2018 Frontiers, Levanon-Spanier et al 1979 Deep Sea Res.)

AC6: Thank you for the references, we included them in our manuscript.

RC7.- Page 5-6: Silicate is measured, mentioned in the results and in many figures/tables (with significant interactions) but nowhere mentioned in the Discussion. If not important, mention briefly in Discussion. Page 6-7, line 20 and 1 (resp.): Was NH4 determined? If not, then you have NOx values, not DIN.

AC7: Thank you for pointing this out. We have changed DIN by NOx through the manuscript. Regarding the lack of discussion about the specific relationship between metabolic rates and silicates, we would like to mention that we did not discuss in detail the relationship between metabolic rates and any of the inorganic nutrients measured, as we aimed to discuss the overall patterns found among all variables (i.e., nutrients, temperature, autotrophic biomass). However, we agree that it would be worth exploring these relationships in detail in future work, perhaps a manuscript mostly centred towards that goal.

RC9.- Page 7, line 10: provide actual depths of PAR measurements. Also, I am confused about the use of 100%, 60-20 and 8-1 as table 1 and 2 give different ranges. Is the data comparable if different depths of sampling were used?

AC9: The reviewer is absolutely right, and we understand the reviewer's confusion. We have modified the text, tables and figures to clarify this point. During our surveys, the samples to quantify planktonic metabolic rates were consistently taken within the first optical depth (zeta), towards the end of the photic layer (3–4.6 zeta), and one intermediate sample either taken where we found the max. Chl-a fluorescence, or in case the Chl-a max was at the surface or the bottom layers, the intermediate sample was collected towards the middle of the euphotic layer (approx. 2.3 zeta). Therefore, our measurements are comparable. We chose the sampling based on the optical depths instead of physical depths (i.e., m) as they are biologically relevant to describe

metabolic processes such as GPP. In Red Sea waters, there is a marked latitudinal and seasonal variability in the apparent optical properties. For example, during March 2018, the vertical attenuation coefficient of downward photosynthetically available radiation (Kd PAR) changed across the basin, from 0.062 m-1 towards the southern region to 0.051 m-1 approx.21° N, while in the northern area, the Kd was 0.059 m-1. Due to this variability, the physical depths ("actual depths") corresponding to the sampled optical depths varies within cruises and regions.

RC10. Page 7: Were samples for metabolic rates filtered? Does the planktonic community include both single and multicellular organisms? Were the optodes adjusted for salinity? A major, potential, flaw in the methods used for metabolic rates is that it appears as though net photosynthesis was measured for 24h. If correct, this includes an approx. A 12-hour period of darkness and thus results in data that cannot be used for calculations for gross photosynthesis, i.e. O2 measurements will be severely lower due to dark respiration. net photosynthesis should have been measured only during daylight and respiration rates should have been measured in 2 phases; during the daytime and during nighttime, so approximately 12:12 h as respiration rates can have a diurnal rhythm. So extrapolating these data to daily rates could result in a wrong estimation of gross photosynthesis Also, how was O2 production data extrapolated to per day? I suggest authors stick to hourly values for oxygen rates. If methods are used correctly, carbon budgets can be calculated using a conversion factor. If net photosynthesis was measured during daylight and respiration for 24 hours, the authors need to state assumptions of the values to their manuscript (potential over- or underestimation of rates)

AC10: We agree with the reviewer that to measure net photosynthesis (i.e., the net organic carbon production in the light), a 24-h incubation can carry potential bias. However, in our study, we are not measuring nor reporting net photosynthetic rates. As clarified in our AC5 and also stated in our introduction and methodology, our goal was to quantify the daily net community production. Therefore, we aimed to estimate the

metabolic contribution not only from the autotrophic community but the entire planktonic community (i.e., the balance between the production and respiration of organic material). The selected methodology to quantify planktonic metabolism is based on the extensively used dark and light method in combination with the Winkler titration method (as mentioned in our reply AC5) and not optodes (as mentioned by the reviewer), in a 24 h incubation period, hence that we report our results as daily rates.

RC12.- Page 9, statistics: Need to be expanded with actual models used.

AC12: We included an extended explanation of the statistical analyses

RC13.- Page 9, line 11: NOx, not DIN.

AC13: We modified the text accordingly

RC14.- Page 10, line 17: 56% of heterotrophs suggests dominance of this trophic strategy

AC14: We modified the text to clarify this point.

RC15.- Page 11, line 4: What models were used to test this?

AC15: We determine if plankton metabolism and nitrate +nitrate were correlated by using a Pearson's correlation. We clarified this point in the new version of the manuscript now between lines 261–262.

RC16.- Page 11, line 8: Which analysis?

AC16: OLS linear regression. This information is now been stated in section 2.4 (statistical analyses) and was in the associated figure (Figure 8).

RC17.- Page 11, line 11: Introduction of this statistical method should be in the appropriate section

AC17: We included this information as suggested in section 2.4.

RC18.- Page 11: How were AE values calculated? - Page 11: AE are presented as

negatives, are they? Next page the authors mention a positive value.

AC18: The procedure to obtain the activation energies were explained in the methods section 2.3 (page 8 between lines 16–21). That been said, we determined the activation energies by fitting an OLS linear regression to the relationship between the natural logarithm of Chl-a specific metabolic rates and the inverse of the absolute temperature. The slopes of these so-called Arrhenius plots represent the average activation energy.

The negative values seen in Figure 10, resulted from the way we plotted the normalised metabolic rates against the inverse of the absolute temperature times the Boltzmann's constant (from 38 to 39.5 eV-1, lower x-axis), the slope of the resulting relationship is negative. However, please note that the relationship with the temperature (upper x-axis) would yield a positive slope if plotted from 20.7 to 32.3 °C.

RC19.- Page 13, line 14: GPP is said to be low, compared to what?

AC19: We clarified this point in the new version of the manuscript. Now in line 328.

RC20.- Page 15, line 18: How was AE standardized to Chl-a?

AC20: We understand the confusion, we clarified this point as we were talking about the AE obtained from the Chl-a normalised GPP.

RC21.- Page 16, line 2: What is "the ocean"?

AC21: We modified this noun, as we meant open oceanic waters.

RC22.- Page 16: Opens with "Surprisingly" and a discussion, then the next paragraph mentions a contradiction that is not surprising. What is the contradiction exactly the authors mean?

AC22: Thank you for highlighting this point, we agree with the reviewer, we deleted this adverb, as it is misleading.

RC23 Page 16, line 6-7: Authors compare results with other references but need to

mention actual values.

AC23: We added the relevant information. The changes can now be tracked between lines 387–390 in the new version of the manuscript.

RC24.- Figure 3: Thickness of the pink or green seems to say something about how significant it is but this is said nowhere. In line with this, the diagonal dark green lines seem to signify extreme significance instead of same variable and thus not tested. DIN is NOx. Are variables tested at different depths than metabolic rates of plankton? If so, how can you relate the 2?

AC24: We agree with the reviewer's comments. Therefore, we decided to modified this graph to present information relevant to the depths we analyse metabolic rates. Also, we changed the type of graph and explained the colour code.

RC25.- Figure 4-6: Lots of white space and hard to see with tiny colored dots anyway. Revise these figures. I suggest to distill from them the most important results you want to show and add the rest to the supplementary section.

AC25: We present this information in the supplementary section.

RC26.- Figure 7: could be mentioned with text in the results section. Suggest moving figure to supplements.

AC26: The information regarding this figure is mentioned in the results, but as it also provides information to derive the GPP threshold we prefer to keep as the main result.

RC27.- Figure 9: Same as Figure 7, B is missing a parenthesis on the y-axis - Figure 10: Same as Figure 7.

AC27: Noted, we corrected the typo. However, we prefer to keep the figure as the main result.

RC28.- Please use continues line numbers for the manuscript

AC28: We followed the format designated by the journal. However, in the new version of the manuscript, the changed to continues lines as suggested by the reviewer.

RC29.- Page1, line 8-9: Please rewrite, it reads as if you want to understand their variability and their present and their future but you want to understand their variability in the present and the future

AC29: Noted, this has been modified as suggested

RC30.- Page 2, line 4-5: Add community

AC30: Done as suggested

RC31.- Page 2, line 11: First mention of NCP, introduce abbreviation.

AC31: The term NCP was mentioned for the first time Page 1 line 14. Therefore, we are not modifying the line.

RC32.- Page 3, line 1: "The Red Sea is a semi-enclosed"

AC32: Noted

RC33.- Page 3, line 3-5: Consider merging this sentence with the previous one

AC33: Done as suggested

RC34.- Page 3, line 9: "throughout the year"

AC34: Noted

RC35.- Page 3, line 10: Delete the dot before the references

AC35: Noted

RC36.- Page 4, line 12: Add "relatively" to "unproductive waters"

AC36: Done as suggested

RC37.- Page 4, line 18: Add "latitudinal gradient" to the sentence

AC37: Done as suggested

RC38.- Page 10, line 8-10: I suggest to start the Results section with this sentence

AC38: thank you for the suggestion but we prefer to keep the sentence as a closing sentence.

RC39.- Page 10, line 16: net autotrophic?

AC39: The sentence was modified accordingly

RC40.- Page 12, line 8: Please stay consistent, use R2.

AC40: Noted

RC41.- Page 13, line 9: Heterotrophic suggest no autotrophs, add "net"

AC41: Done as suggested

RC42.- Page 15, line 6-9: Please rewrite.

AC42: Modified as suggested

RC43.- Page 16, line 4: Add i.e. or parentheses after 2.5 _C

AC43: Changed as suggested.

RC44.- Page 19, line 6: Heterotrophic

AC44: Noted

RC45.- Figure A1: Add axis titles to every part of the figure, having double axes without titles is confusing, especially since the 27 _N axis title (Temperature) is not on any axis.

AC45: Changed as suggested

RC46.- Table 1: Add Silicate to the table description. Also, it is unclear which header belongs to which environmental variable. Also, I fail to see the benefit of the min and max values

AC46: We modified the table accordingly.

RC47.- Present data as mean +/- SE

AC47: Noted

RC28.- Table 2: N does not need decimals

AC48: Noted

RC29.- What does "rank" mean? % PAR differs from Table 1

AC29: Noted, we modified the table.

RC30.- Table 3: Upper part are, what seems to be, Pearson rank coefficients, not the units given in the description. The lower part seems to be p-values, mention this in the description. A hyphen is not the same as a blanc.

AC30: Thank you for pointing this out. It is indeed a correlation matrix, with the r coefficients and p-values. We decided to summarised the table in a new figure

References:

Agusti, S., J. Martinez-Ayala, A. Regaudie-de-Gioux, and C. M. Duarte. 2017. Oligotrophication and Metabolic Slowing-Down of a NW Mediterranean Coastal Ecosystem. Frontiers in Marine Science 4.

Carpenter, J. H. 1965. The accuracy of the Winkler method for dissolved oxygen analysis. Limnology and Oceanography 10: 135-140.

Ducklow, H. W., and S. C. Doney. 2013. What is the metabolic state of the oligotrophic ocean? A debate. Ann Rev Mar Sci 5: 525-533.

Duarte, C. M., and S. AgustÄśÌĄ. 1998. The CO2 balance of unproductive aquatic ecosystems. Science 281: 234-236.

Duarte, C. M., and A. Regaudie-de-Gioux. 2009. Thresholds of gross primary production for the metabolic balance of marine planktonic communities. Limnology and Oceanography 54: 1015-1022.

Falkowski, P. G., and J. A. Raven. 2007. Aquatic photosynthesis, 2nd ed. Princeton University Press.

Gaarder, T., and H. H. Gran. 1927. Investigations of the production of plankton in the Oslo Fjord, p. 1-48. In C. P. I. P. L. E. d. l. Mer [ed.], Repprts et Prcès-Verbaux des Réunions.

García-Martín, E. E. and others 2017. Seasonal changes in plankton respiration and bacterial metabolism in a temperate shelf sea. Progress in Oceanography.

Gran, H. H. 1927. The production of plankton in the coastal waters off Bergen: March-April 1922. In R. o. N. F. a. M. Investigations [ed.].

López-Urrutia, Á., E. San Martin, R. P. Harris, and X. Irigoien. 2006. Scaling the metabolic balance of the oceans. Proceedings of the National Academy of Sciences 103: 8739-8744.

McIntire, C. D., R. L. Garrison, H. K. Phinney, and C. E. Warren. 1964. PRIMARY PRODUCTION IN LABORATORY STREAMS 1,2. Limnology and Oceanography 9: 92-102.

Owens, L. J., and G. W. Crumptom. 1995. Primary production and light dynamics in an upper Mississippi River Backwater. Regulated Rivers: Research & Management 11: 185-192.

Regaudie-de-Gioux, A., and C. M. Duarte. 2012. Temperature dependence of plank-tonic metabolism in the ocean. Global Biogeochemical Cycles 26.

Regaudie-de-Gioux, A., and C. M. Duarte. 2013. Global patterns in oceanic planktonic metabolism. Limnology and Oceanography 58: 977-986.

Robinson, C., and P. J. l. B. Williams. 1999. Plankton net community production and

dark respiration in the Arabian Sea during September 1994. Deep Sea Research Part II: Topical Studies in Oceanography 46: 745-765.

Robinson, C., and P. I. B. Williams. 2005. Respiration and its measurement in surface marine waters. Respiration in aquatic ecosystems: 147-180.

Williams, P., R. C. T. Raine, and J. R. Bryan. 1979. Agreement between the c-14 and oxygen methods of measuring phytoplankton production-reassessment of the photosynthetic quotient. Oceanologica Acta 2: 411-416.

Williams, P. I. B., K. Heinemann, J. Marra, and D. Purdie. 1983. Comparison of 14C and O2 measurements of phytoplankton production in oligotrophic waters. Nature 305: 49.

---

## Author Comment (AC2) · 25 Feb 2019

Reviewer 2 general comments: The authors quantified plankton metabolic rates along the Red Sea. They have shown that Chla and planton community metabolism (GPP and CR) increase with temperature. Contrary to previous results they have observed a higher Activation Energy for GPP than for CR showing a positive relationship between NCP and Temperature. These results have been explained by the authors as a consequence of the high nutrient availability in warmer waters and the lack of external organic carbon sources to sustain a heterotrophic metabolism constraining the CR. The dataset are very interesting and merit been published, however, the way how the results have been presented, the lack of statistical analyses and the methodology proposed are not the most suitable to achieve the main goal proposed in the manuscript.

Therefore, I consider the ms still needs major revision in order to be published and providing the authors follow the reviewers recommendations.

Reply to reviewer general comments:

Thank you for your comments and suggestions. We have addressed the changes and recommendations of the reviewer and in the following section is a detailed answer to each of the points made by R2. First, we totally agree with the reviewer that a detailed description of the statistical analyses performed was missing in the methodology section. We have included a detailed description of the statistical analyses in a new section (2.4), and this change can be tracked now between lines 176–189 in the latest version of the manuscript. Regarding the primary concern of the reviewer 2, which was the methodological approach we used to quantify planktonic metabolic rates, we think, as explained to R1, there is a misunderstanding. The methodology used to quantify planktonic metabolism is based on the extensively used dark and light method (in combination with the Winkler titration method). The reviewer indicated that the methodology used was not suitable, and suggested that a shorter incubation period (6 –12 h) was more appropriate to quantify NCP. We want to point out that NCP represents the organic matter remaining after consumption of the GPP through respiration by plants (autotrophs), microbes (either autotrophs or heterotrophs), and animals (heterotrophs) (Ducklow and Doney 2013), and to account those process, the standard incubation time for in vitro incubations is 24 h. This incubation length is needed because contrary to photosynthesis that can be resolved during daylight, the losses due to respiration (which are necessary to define NCP) also occurs at night.

Reply to specific comments:

Reviewer Comments (RC) Author Reply (AC)

RC1: First, according to the title and the abstract the authors consider as drivers of the plankton community metabolism in the Red Sea, the Chla and temperature. However, other important parameters such as, temporal and spatial variability, salinity and nutrients seem to govern the plankton community metabolism within this particular ecosystem and are not included in the abstract. Therefore, this lack of agreement between the ms, the consclusion and the abstract. is confusing. In my opinion, there is a large floor in the experimental design proposed and it is difficult to resolve.

AC1: We appreciate the reviewers' comment, and agree that the abstract highlighted our main findings and did not detail all the results. The abstract was indeed mostly oriented to the effect of temperature and nutrients availability on metabolic rates as we found that those were main controlling drivers. That was consistently explained on our results, discussion and conclusion, therefore we do not find disagreement in our statements.

RC2: All samples included the deepest ones have been incubated on deck with surface water. During some of the surveys there is an important thermal variability. The authors have attempted to mitigate the issue by including just those samples above the thermocline. However, Material and Methods mention that changes in temperature and PAR in the incubation tanks were recorded with HOBO data loggers. Therefore, those data should be shown in a table in order to select objectively the samples for the analyses. Hence, eliminating those samples that register thermal differences above 2_C with the in situ temperatures. In addition, samples adapted to cool temperatures such as those at the bottom will respond more drastically to artificial increments of temperature than surface ones (for example. Apple et al. 2006. AME. 43: 243–254) resulting in erroneous conclusions. Therefore, Figure A1 is important and should be included in the Ms.

AC2: Thank you for your comment and reference, we have moved Figure A1 into the text results section and now is presented as the main figure.

RC3: Other figures such as 4-6 do not show crucial information in the current format. Figure 3 and Table 3 to me are redundant.

AC3: We moved figures 4–6 as supplementary information. Figure 3 and table 3 are

complementary as table 3 indicates the correlation between metabolic rates environmental variables.

RC4: The paragraph 10-15 page 6 the authors should indicate if samples were collected before sunrise (to avoid any light on the samples) and if the incubation started at the sunrise to estimate the full light period. The authors say, the samples were colleted between 7 to 9 and to me this sounds very late to incubate and obtain the full light period nor precisely.

AC4: Samples were incubated for 24-h covering an entire light-dark period.

RC5: In The net community metabolism..... page 7, NCP should be estimated during the light period (NCP 6 to 12 hours).

AC5: We believe that there is a misunderstanding regarding the process we measured. The reviewer comment seemed to suggest that our work was focused on primary production, which is performed by the photosynthetic components of the plankton community during the daytime and that has gross and net components (as phytoplankton excrete and respire carbon). However, our paper focuses on the entire plankton community, both photosynthetic and heterotrophic (e.g. bacteria), where the net community production (NCP) represents the organic matter remaining after consumption of the GPP through respiration by plants (autotrophs), microbes (either autotrophs or heterotrophs), and animals (heterotrophs) (Ducklow and Doney 2013). Studies that focus on the photosynthetic component of the plankton community (e.g. Net Primary Production, NPP) report values for the daylight period only. Whereas studies, such as this, report (24 h) rates. For instance, published synthesis of community metabolism rates report values per day (24 h), e.g., Robinson and Williams 2005, Regaudie-de-Gioux 2012, 2013). The use of 24 h to report rates is justified as the metabolic budget need be resolved over 24 h to be completed, for photosynthesis this does not proceed at night, but respiration, which is necessary to define net community production, occurs at night as well.

RC6: The authors should show, the variation coefficient of the pool data and also the original CR, NCP and GPP data including their SE.

AC6: We already presented the mean and SE of our metabolism measurements in Table 3, and now we have included CV.

RC7: Because, In these oligrotophic areas the metabolic rates are very low and can be difficult to detect. Therefore, the methodology needs to be very precise in the processes of filling, incubating and fixing the bottles.

AC7: We agree with the reviewer, however, the information about the filling and all special cares during the sampling are already detailed in methods section 2.3. Now between lines 136–155.

RC8: The paragraph 20 in page 8 It should be indicated the Arrhenius plots the authors mention.

AC8: The Arrhenius plots described in the methods section 2.3 (P8, between lines 13 and 20) where already shown and explained in the results section 3.3, when we described the response of planktonic metabolism and temperature. Now between lines 305–315.

RC9: The paragraph 10 in page 10 should be transfered from the Results to the Discussion.

AC9: The sentence in P10, between lines 8–10 is a closing statement with the main results shown in previous paragraphs, and there we are not discussing any results. Therefore, we prefer to keep it as it is.

RC10: And also the first paragraph of the 3.2 Variability of plankton : : :. Is already mentioned in M and M.

AC10: We modified the text as suggested

RC11: The paragraph 10 in page 13 There are lots of references within oligotrophic

areas very intersting and different to the authors ones that the authors should also include in the MS.

Thank you for pointing this out, we added new relevant references.

RC12: Figure 1. The name of the KAUST is excessive. I would use just one larger map with different colours or shapes to show the stations at each survey or season.

AC12: We decided to not modify the figure as some of the stations are sampled on the same location more than twice, and different shapes or points will be overlapped.

RC13: Figure 2. I consider in this figure is difficult to detect the thermocline and the vertical profiles of Chla and salinity. I consider that nutrient profiles should also included

AC13: It is not possible to determine the depth of the thermocline in Figure 2, and it is not intended to do so. The figure summarises the main characteristics of water column properties at different optical depths. Perhaps the reviewer meant figure 1A?. If so, nitrate+nitrate concentration is plotted.

RC14: Figure 8, 9 and 10. To test one of the main conclusions, if AE is higher for GPP than for CR, authors should test statistically if the slopes are different. I would test also the slopes for the figures 9 and 10 explaining the consequences of the statistical differences in the cases observed

AC14: We did perform a test (an analysis of covariance, ANCOVA) to compare the regression lines and test if the interaction of the metabolic rates with the inverse of temperature was significantly different from zero (meaning that the effect of temperature on metabolism depends on the level (e.g., season). The results of the analyses were described between lines 1–6 (page 12) and discussed in section 4.2. In the new version of the manuscript, we have included a new section (2.4) detailing all the statistical analyses. Methods section 2.4, this can be tracked between lines 176–189.

RC15: In the figure 9, the RMA analyses have been included but it is not necesary in this case because temperature is not a rate. In addition, the authors have not explained

when the RMA or OLS should be used in M and M.

AC15: We agree with the reviewer that it is not necessary to provide the results of the RMA analyses. Therefore, we decided to remove from our manuscript.

---

## Author Response (AR2)

June 27, 2019

Dear Dr Ciavatta,
Associate Editor, Biogeosciences

Please accept our sincere apologies for the mismatch found between the lines indicated in our reply and the manuscript changes, here we are re-submitting our reply and the manuscript "*Rates and drivers of Red Sea plankton community metabolism*" with tracked changes. We have corrected the issue, and now the lines indicating the changes in the new manuscript can be found in the following section and our reply to the reviewers:

1. About the different variables used to explain the changes of planktonic metabolism in the Red Sea (i.e., latitude, water depth and seasonality) and reviewers' suggestion to rewrite the narrative of the manuscript by focussing on one or two variables.

   *Reply*
   We had modified the text and had mainly focused on the latitudinal and seasonal variability of planktonic metabolism and their relationship with temperature and nutrient availability (both tightly related to the latitudinal gradient that characterised the Red Sea). Although changes throughout the water column are relevant, we agreed with the reviewer that it added an unnecessary complexity to the text.

   *Action*
   We re-wrote most of the results section (now between lines 254–387 of the new version of the manuscript which includes the track changes), and modified the figures so that Figure 2 describes the latitudinal gradient of variables only in surface waters. Figure 3: we modify the format and the data presented, so we only included data within the photic layer. We removed Figures 4–5, and Figures 1A and 2A. We also removed Table 1, and Table 2. The information from Table 2 is now presented as a figure (now Figure 5) with the overall information and with all the data plotted. The changes in the results are in agreement with changes in the narrative of the discussion section. The main changes in the discussion section can be tracked between lines 447–463 and between lines 550–559.

2. About the methodology chosen and length of the incubations.

   *Reply*
   The Winkler titration method used in this study, is a methodology extensively used in studies that aim to measure autotrophic planktonic photosynthetic rates (i.e. gross primary production rates) and respiration rates derived from both the autotrophic and heterotrophic plankton community (see e.g., Williams et al., 1979; Duarte and Agustí, 1998; Bender et al., 1999; Robinson and Williams, 1999; Ducklow et al., 2000; Serret et al., 2001; Robinson et al., 2002; Serret et al., 2009; García-Martín et al., 2017). The selected incubation time allows to fully take into account diel changes of both metabolic rates, as community respiration rates occur during day and night.

   *Action*

We have clarified this point in the methods section 2.3 (between lines 166–174). In addition, we have ensured that the sampling, filling and fixing of the samples is described in sufficient detail. We also clarified how we calculated gross primary production, community respiration and net community production (which is the balance between the autotrophic and heterotrophic metabolism, and not only the result of photosynthetic activity). These calculations can be seen between lines 206 and 216.

3.  About the lack of statistical analyses.

*Reply*
We agreed with the reviewers that a section with a detailed description of all the statistical analyses was missing, and that the statistical analyses mentioned in the results were not properly clarified, hence the reviewers were under the impression that we did not perform any statistics to support our conclusions.

*Action*
We have included a detailed description of all the statistical analyses done (see section 2.4 between lines 227–253). In addition, the results of each analysis are detailed whenever they are presented. See e.g. line 354 or line 383.

4.  We also detected some typos and errors in three figures and one table that have been rectified, yet neither of those changed our main conclusions in any way.

Figures:
- Figure 8C (now Figure 4G) $R^2$ is 0.38 not 0.39 and, $R^2$ in Figure 8I is 0.16 not 0.17
- Figure 7A (now Figure 6A), the intercept is 0.73 not 0.74
- Figure 9B (now Figure 7B), the intercept is -0.73 not -0.72

The detailed answer to the reviewers with the updated line numbers is presented in the following section.

I would be grateful if this newly revised version of the manuscript was considered for publication in Biogeosciences.

Sincerely,
Daffne C. López-Sandoval
Red Sea Research Center
King Abdullah University of Science and Technology (KAUST)
Thuwal, Jeddah, 23955-6900, Kingdom of Saudi Arabia
Phone (+966) 12 808 2659

**Reply to reviewers**

**Reviewer 1**

**2.1.1 General comments**

The authors describe a dataset of environmental variables related to the metabolism of planktonic communities along a depth and latitudinal gradient in a seasonal resolution in the Red Sea. The authors conclude that gross primary production relates positively to sea surface temperature and nutrient availability. The dataset is extensive, and the research questions (for this part of the Red Sea), to my knowledge, are novel and worthy of publication. The abstract is clear and reads well, but shows a different narrative than the rest of the manuscript. Thus, I suggest for the authors to consider rewriting the manuscript. As mentioned in the author contributions, the manuscript is written by several people and this is noticeable (see specific comments). The abstract mentions the latitudinal gradient but the ms introduces two more variables, i.e. depth and seasonality. While interesting variables, they make the story confusing at times and harder to disentangle the story the authors want to tell
(according to the abstract). Concerns about the methods used are mentioned in the specific comments and need to be addressed first. Proper description of statistical analyses is lacking.
My recommendation is that the ms needs major revisions, but only if methodological concerns can be addressed adequately. Then, I suggest a complete overhaul of the manuscripts narrative by focusing on 1 or 2 of the 3 major variables (latitude, water depth and seasonality) and stick with these in the entire narrative. Also, there needs to be a clear description of used statistics in the M&M section and figures and tables should be cut back and/or improved. Consistency in the presentation of the results (including the statistics) and the use of abbreviations (as well as changing them) is recommended

*Reply to general comments*

We sincerely appreciate the thorough revision, comments and the time devoted to review our manuscript in such a rigorous manner, as it helped us to greatly improve our manuscript. We have addressed all the reviewer's comments, and significant changes were done to the manuscript, figures and tables. Also, we have included a detailed description of the statistical analyses.

*Action*

The changes can be tracked in the new version of the manuscript in the Results section between lines 254–387 of the new version of the manuscript with track changes. We modified the figures according to the new narrative of the results as such that Figure 2 now only describes the latitudinal gradient of variables in surface waters. In Figure 3, we modified the format and the data presented, so now we only focus on data within the photic layer. We removed Figures 4–5, Figure 1A and 2A, as well as Table 1 and Table 2. The information from Table 2 is now presented as a figure (now Figure 5) with the overall information and with all the data plotted. The changes in the results are in agreement with changes in the narrative of the discussion section. The main changes in the discussion section 447–463 and between lines 550–559.

**2.1.2 Reply to specific comments**

**Reviewer comment 1**

Title: Says Red Sea but Gulfs are not included.

*Reply*

We agree with the reviewer that our study does not include data from the Gulf of Aqaba nor does it include the western half of the Red Sea. However, this detail is carefully described in the Introduction (line 123) and Methods sections (between lines 139–140) and in Figure 1, so the reader will not be misled.

*Action*

We, preferred to leave the title as it is, since we provided sufficient detail in the manuscript on which areas were sampled.

**Reviewer comment 2**

Abstract: Line 10: Mentioning "Low productive waters" immediately brings down the importance of the story.

*Action*

We modify the text accordingly and deleted "low productive waters" (see line 10)

**Reviewer comment 3**

The first paragraph is loaded with self-referencing while many others are not or less.

*Reply*

There is no intent to load the paper unnecessarily with self-references, but it turns out that the co-authors of this paper have published a large number of relevant articles on the topic. We have now, however, added additional works, by other researchers.

*Action*

These new references can be tracked between lines 39–44 and between 103–104.

**Reviewer comment 4**

Page 2, line 4-5 and 11: Introduce abbreviations once (see technical corrections) and use them consistently throughout the ms.

*Reply*

Thank you for your comment. We have modified the text accordingly.

*Action*

Abbreviations were introduced between lines 13 and 14 and subsequently used consistently.

**Reviewer comment 5**

Page 2: The abbreviations of GPP, CR and NCP are presented with units of daily oxygen produced or used. However, these abbreviations are normally used for daily production and use of carbon. I suggest the authors change the abbreviations for these processes and/or use a conversion factor to present daily carbon production and use.

*Reply*

This and other comments from the reviewer, led us to believe that there is a misunderstanding as they seem to suggest that our work was focused on primary production, which is performed by the photosynthetic components of the plankton community during the daytime and that has gross and net components (as phytoplankton excrete and respire carbon). However, our paper focuses on the entire plankton community, both photosynthetic and heterotrophic (e.g. bacteria), where the net community production (NCP) represents the organic matter remaining after consumption of the GPP through respiration by plants (autotrophs), microbes (either autotrophs or heterotrophs), and animals (heterotrophs) (Ducklow and Doney 2013). Hence, it is the balance of the photosynthetic processes (GPP) and respiratory activity of the entire plankton community (not just phytoplankton) measured during 24 h periods. Studies that focus on the photosynthetic component of the plankton community (e.g. Net Primary Production, NPP) report values for the daylight period only whereas studies such as this, report (24 h) rates. For instance, published syntheses of community metabolism rates report values per day (24 h), e.g., Robinson and Williams 2005, Regaudie-de-Gioux 2012, 2013).

Reports of daily GPP, NCP and CR rates in oxygen units has remained the case since Haaken H. Gran used the light and dark bottle method in combination with the Winkler method to report community metabolic rates (Gaarder and Gran 1927; Gran 1927). The reviewer can find other examples in Mclntire et al. 1964; Williams et al. 1979; Williams et al. 1983; Owens and Crumptom 1995; Robinson and Williams 1999; López-Urrutia et al. 2006; Stanley et al. 2010; García-Martín et al. 2017. The use of 24 h to report rates is indeed not just a tradition but is justified as the metabolic budget needs be resolved over 24 h to be completed. For photosynthesis this does not apply as it is a light-dependent process, but respiration, which is necessary to define net community production, occurs at night as well. Moreover, all of the above papers report rates based on oxygen units since this is the property measured, and converting these to C requires some assumptions, such as a theoretical PQ value. However, to allow comparisons with another component of carbon cycling, we have provided an estimate of what, for example, the mean GPP reported here represents in terms of carbon production, assuming a PQ=1.

*Action*

We have clarified this point in the Methods section 2.3 (between lines 166–174). In addition, we clarified how we calculated gross primary production, community respiration and net community production (which is the balance between the autotrophic and heterotrophic metabolism, and not only the result of photosynthetic activity). These calculations can be seen between lines 206 and 216. The average GPP rates in carbon units can be tracked between lines 302–303. We have also included the equivalent of CR in carbon units (assuming a RQ=1) in line 305.

**Reviewer comment 6**
Page 4, line 7-8: There are plenty of references that describe metabolism in the northern part of the Red Sea (e.g. Rahav et al. 2015 MEPS, Tilstra et al 2018 Frontiers, Levanon-Spanier et al 1979 Deep Sea Res.)

*Reply*

Thank you for the references, we have included them in our manuscript.

*Action*

Please find the suggested references and others more between lines 103–104.

**Reviewer comment 7**

Page 5-6: Silicate is measured, mentioned in the results and in many figures/tables (with significant interactions) but nowhere mentioned in the Discussion. If not important, mention briefly in Discussion. Page 6-7, line 20 and 1 (resp.): Was NH4 determined? If not, then you have NOx values, not DIN.

*Reply*

Thank you for pointing this out. Regarding the lack of discussion about the specific relationship between metabolic rates and silicates, we would like to mention that we did not discuss in detail the relationship between metabolic rates and any of the inorganic nutrients measured, as we aimed to discuss the overall patterns found among all variables (i.e., nutrients, temperature, autotrophic biomass).

*Action*

We have replaced DIN with NOx throughout the manuscript. Please see lines 261 and 306, as well as Figure 3, and Table 1.

**Reviewer comment 8**

Page 7, line 10: provide actual depths of PAR measurements. Also, I am confused about the use of 100%, 60-20 and 8-1 as table 1 and 2 give different ranges. Is the data comparable if different depths of sampling were used?

*Reply*

The reviewer is absolutely right, and we understand the reviewer's confusion. During our surveys, the samples to quantify planktonic metabolic rates were consistently taken within the first optical depth ($\zeta$), towards the bottom of the photic layer, and one intermediate sample was either taken where we found the max. Chl-a fluorescence, or in case the Chl-a max was at the surface or the bottom layers, the intermediate sample was collected towards the middle of the euphotic layer (approx. 2.3 $\zeta$). Therefore, our measurements are comparable. We chose the sampling based on the optical depths instead of physical depths (i.e., depth in m) as they are biologically relevant to describe metabolic processes such as GPP.

*Action*

This has been clarified in the manuscript between lines 175–188

**Reviewer comment 9**

Page 7: Were samples for metabolic rates filtered? Does the planktonic community include both single and multicellular organisms? Were the optodes adjusted for salinity? A major, potential, flaw in the methods used for metabolic rates is that it appears as though net photosynthesis was measured for 24h. If correct, this includes an approx. 12-hour period of darkness and thus results in data that cannot be used for calculations for gross photosynthesis, i.e. O2 measurements will be severely lower due to dark respiration. net photosynthesis should have been measured only during daylight and respiration rates should have been measured in 2 phases; during the daytime and during nighttime, so approximately 12:12 h as respiration rates can have a diurnal rhythm. So extrapolating these data to daily rates could result in a wrong estimation of gross photosynthesis Also, how was O2 production data extrapolated to per day? I suggest authors stick to hourly values for oxygen rates. If methods are used correctly, carbon budgets can be calculated using a conversion factor. If net photosynthesis was measured during daylight and respiration for 24 hours, the authors need to state assumptions of the values to their manuscript (potential over- or underestimation of rates)

*Reply*

We agree with the reviewer that to measure net photosynthesis (i.e., the net organic carbon production in the light), a 24-h incubation can carry potential bias. However, in our study, we are neither measuring nor reporting net photosynthetic rates. As clarified in our reply to specific comments (5) and also stated in our introduction and methodology, our goal was to quantify the daily net community production. Therefore, we aimed to estimate the metabolic contribution not only from the autotrophic community but the entire planktonic community (i.e., the balance between the production and respiration of organic material). The selected methodology to quantify planktonic metabolism is based on the extensively used dark and light method in combination with the Winkler titration method (as mentioned in our reply AC5) and not optodes (as mentioned by the reviewer), in a 24 h incubation period, hence we report our results as daily rates.

*Action*

We have clarified why we used this methodology, and supported with references that the methodology chosen is largely used for the goals of this manuscript. These points can be seen in the methods section 2.3 (between lines 166–174)

**Reviewer comment 10**
Page 9, statistics: Need to be expanded with actual models used.

*Reply*

We agree with the reviewer that a section with a detailed description of all the statistical analyses was missing.

*Action:*

We have included a detailed description of all the statistical analyses done (see section 2.4 between lines 227–253) and in addition the results of each analysis are detailed whenever they are presented. See e.g. line 354 or line 383.

**Reviewer comment 11**
Page 9, line 11: NOx, not DIN.

*Action*

We modified the text accordingly, as seen now in line 261.

**Reviewer comment 12**
Page 10, line 17: 56% of heterotrophs suggests dominance of this trophic strategy

*Reply*

We agreed with the reviewer

*Action*

We have modified the results section and we no longer describe by depths the autotrophic or heterotrophic status of the communities

**Reviewer comment 13**

Page 11, line 4: What models were used to test this?

*Reply*

We determined if plankton metabolism and nitrite and nitrate were correlated by using a Pearson's correlation

*Action*

We re-analysed the data and the results are shown between lines 306–309

**Reviewer comment 14**

Page 11, line 8: Which analysis?

*Reply*

We evaluated the relationship between GPP with CR and NCP by performing an OLS linear regression.

*Action*

This information has now been clarified in the statistical analyses section (between lines 234–249) and in the associated figure (Figure 6).

**Reviewer comment 15**

Page 11, line 11: Introduction of this statistical method should be in the appropriate section

*Reply*

We agree with the reviewer and have included it in the statistical analyses.

*Action*

Please refer to line 234–249, the rationale behind the Arrhenius plots was already described in section 2.3. This information can now also be found between lines 218–226.

**Reviewer comment 16**

Page 11: How were AE values calculated? - Page 11: AE are presented as negatives, are they? Next page the authors mention a positive value.

*Reply*

The procedure to obtain the activation energies were explained in the methods section 2.3 (page 8 between lines 16–21, now between lines 218–226). That being said, we determined the activation energies by fitting an OLS linear regression to the relationship between the natural logarithm of Chl-a specific metabolic rates and the inverse of the absolute temperature.

The slopes of these so-called Arrhenius plots represent the average activation energy. The negative values seen in Figure 10 (now figures 7 and 8), resulted from the way we plotted the normalised metabolic rates against the inverse of the absolute temperature multiplied by the Boltzmann's constant (from 38 to 39.5 $eV^{-1}$, lower x-axis), the slope of the resulting relationship is negative. However, please note that the relationship with temperature (upper x-axis) would yield a positive slope if plotted from 20.7 to 32.3 ºC.

**Reviewer comment 17**
Page 13, line 14: GPP is said to be low, compared to what?

*Action*

We clarified this point in the new version of the manuscript, please see in line 483.

**Reviewer comment 18**
Page 15, line 18: How was AE standardized to Chl-a?

*Reply*

We understand the confusion; the sentence was poorly written. We aimed to described the activation energies obtained from the relationship between the Chl-a specific GPP data and temperature.

*Action*

 We completely changed this section of the discussion, please refer to lines 550–559.

**Reviewer comment 19**
Page 16, line 2: What is "the ocean"?

*Action*

We modified this noun, as we meant open oceanic waters. Please see lines 556–559.

**Reviewer comment 20**
Page 16: Opens with "Surprisingly" and a discussion, then the next paragraph mentions a contradiction that is not surprising. What is the contradiction exactly the authors mean?

*Reply*

AC22: Thank you for highlighting this point

*Action*

We deleted this adverb as it is misleading

**Reviewer comment 21**
RC23 Page 16, line 6-7: Authors compare results with other references but need to mention
actual values.

*Reply*

We added the requested information

*Action*

The changes can now be tracked between lines Please see lines 556–559.

**Reviewer comment 22**

Figure 3: Thickness of the pink or green seems to say something about how significant it is but this is said nowhere. In line with this, the diagonal dark green lines seem to signify extreme significance instead of same variable and thus not tested. DIN is NOx. Are variables tested at different depths than metabolic rates of plankton? If so, how can you relate the 2?

*Reply*

We agree with the reviewer's comments.

*Action*

We modified this graph to present information relevant to the depths where we analysed metabolic rates (i.e., only photic layer). In addition, we modified the way the results are presented, and we explained the color code in figure 3.

**Reviewer comment 23**

Figure 4-6: Lots of white space and hard to see with tiny colored dots anyway. Revise these figures. I suggest to distill from them the most important results you want to show and add the rest to the supplementary section.

*Action*

We decided to remove the figures in this new version of the manuscript

**Reviewer comment 24**

Figure 7: could be mentioned with text in the results section. Suggest moving figure to supplements.

*Reply*

The information regarding this figure is mentioned in the results, but as it also provides information to derive the GPP threshold, we prefer to keep it.

**Reviewer comment 25**

Figure 9: Same as Figure 7, B is missing a parenthesis on the y-axis - Figure 10: Same as Figure 7.

*Reply*

Since figure 9 and 10 are both explaining one of the main results (i.e. the metabolic response of metabolic rates to temperature) we prefer to keep them as main figures.

*Action*

We added the missing parenthesis to the y-axis of the figures.

**Reviewer comment 26**

Please use continues line numbers for the manuscript

*Reply*

We followed the format and template designated by the journal.

*Action*

However, in the new version of the manuscript we changed to continuous lines as suggested by the reviewer.

**Reviewer comment 27**
Page1, line 8-9: Please rewrite, it reads as if you want to understand their variability and their present and their future but you want to understand their variability in the present and the future

*Action*

This has been modified as suggested, please refer to line 9

**Reviewer comment 28**
Page 2, line 4-5: Add community

*Action*

Done as suggested, please see in line 38

**Reviewer comment 29**
Page 2, line 11: First mention of NCP, introduce abbreviation.

*Reply*

NCP was mentioned for the first time on Page 1 line 14

**Reviewer comment 30**
Page 3, line 1: "The Red Sea is a semi-enclosed"

*Action*

We modified the text, please see line 67

**Reviewer comment 31**
Page 3, line 3-5: Consider merging this sentence with the previous one

*Action*

Please see the changes between lines 68–70

**Reviewer comment 32**
Page 3, line 9: "throughout the year"

*Action*

Changed as suggested, see line 75

**Reviewer comment 33**
Page 3, line 10: Delete the dot before the references

*Action*

Noted, see line 76

**Reviewer comment 33**
Page 4, line 12: Add "relatively" to "unproductive waters"

*Action*

Done as suggested, see line 107

**Reviewer comment 34**
Page 4, line 18: Add "latitudinal gradient" to the sentence

*Action*

Done as suggested, see line 112

**Reviewer comment 35**
Page 10, line 8-10: I suggest to start the Results section with this sentence

*Reply*

Thank you for the suggestion but we prefer to keep the sentence as a closing sentence

**Reviewer comment 36**
Page 10, line 16: net autotrophic?

*Reply*

This entire section of the discussion was modified

**Reviewer comment 37**
Page 12, line 8: Please stay consistent, use R2.

*Reply*

This section was modified

**Reviewer comment 38**
Page 13, line 9: Heterotrophic suggest no autotrophs, add "net"

*Reply*

This section was modified

**Reviewer comment 39**
Page 15, line 6-9: Please rewrite.

*Reply*

Modified as indicated, please see lines 540–544

**Reviewer comment 40**
Page 16, line 4: Add i.e. or parentheses after 2.5 _C

*Action*

We modified the text in the section but indicated the range inside a parenthesis (line 554)

**Reviewer comment 41**
Page 19, line 6: Heterotrophic

*Action*

Typographical error corrected, see line 666

**Reviewer comment 42**
Figure A1: Add axis titles to every part of the figure, having double axes without titles is confusing, especially since the 27 _N axis title (Temperature) is not on any axis.

*Action*

We are not including this figure in the new version of the manuscript

**Reviewer comment 43**
Table 1: Add Silicate to the table description. Also, it is unclear which header belongs to which environmental variable. Also, I fail to see the benefit of the min and max values

*Action*

We are not including this table in the new version of the manuscript

**Reviewer comment 44**
Present data as mean +/- SE

*Action*

We clarified this point in line 253. Whenever we presented mean values, we added the standard error of the mean

**Reviewer comment 45**
Table 2: N does not need decimals

*Action*

We are not including this table in the new version of the manuscript

**Reviewer comment 46**
What does "rank" mean? % PAR differs from Table 1

*Action*

We are not including this table in the new version of the manuscript. Data from table 2 are now being presented in a figure (new figure 5) with the overall information by seasons, but with all the data points included in the study.

**Reviewer comment 46**

Table 3: Upper part are, what seems to be, Pearson rank coefficients, not the units given in the description. The lower part seems to be p-values, mention this in the description. A hyphen is not the same as a blanc.

*Reply*

Thank you for pointing this out. It is indeed a correlation matrix.

*Action*

We modified the table (now Table 1) and only presented the correlation of metabolic rates with main explanatory variables.

**Reviewer 2**

**2.2.1 General comments**

The authors quantified plankton metabolic rates along the Red Sea. They have shown that Chla and planton community metabolism (GPP and CR) increase with temperature. Contrary to previous results they have observed a higher Activation Energy for GPP than for CR showing a positive relationship between NCP and Temperature. These results have been explained by the authors as a consequence of the high nutrient availability in warmer waters and the lack of external organic carbon sources to sustain a heterotrophic metabolism constraining the CR.

The dataset are very interesting and merit been published, however, the way how the results have been presented, the lack of statistical analyses and the methodology proposed are not the most suitable to achieve the main goal proposed in the manuscript. Therefore, I consider the ms still needs major revision in order to be published and providing the authors follow the reviewers recommendations.

**2.2.2 Reply to general comments**

*Reply*

Thank you for your comments and suggestions. We have addressed the changes and recommendations of the reviewer and provide a detailed answer to each of the points made in the following section. First, we completely agree with the reviewer that a detailed description of the statistical analyses performed was missing in the methodology section. We have included a detailed description of the statistical analyses in a new section (2.4), and this change can be tracked now between lines 227–253 in the latest version of the manuscript. Regarding the primary concern of Reviewer 2, which was also pointed out by the first referee, related to the methodological approach we used to quantify planktonic metabolic rates, we think, there is a misunderstanding. The methodology used to quantify planktonic metabolism is based on the extensively-used dark and light method (in combination with the Winkler titration method). The reviewer indicated that the methodology used was not suitable, and suggested that a shorter incubation period (6–12 h) was more appropriate to quantify NCP. We want to point out that NCP represents the organic matter remaining after consumption of the GPP through respiration by plants (autotrophs), microbes (either autotrophs or heterotrophs), and animals (heterotrophs) (Ducklow and Doney 2013), and to account for those process, the standard incubation time for *in vitro* incubations is 24 h. This incubation length is needed because contrary to photosynthesis, which can be resolved during daylight, the losses due to respiration (which are necessary to define NCP) also occurs at night.

*Action*

We have clarified this point in the methods section 2.3 (between lines 166–174). In addition, we made sure that a detailed description of the sampling, filling and fixing of the samples was provided. We also clarified how we calculated gross primary production, the community respiration and the net community production (which is the balance between the autotrophic and heterotrophic metabolism, and not only the result of photosynthetic activity). These calculations can be seen between lines 206 and 216.

**2.2.2 Specific comments**

**Reviewer comment 1**
First, according to the title and the abstract the authors consider as drivers of the plankton community metabolism in the Red Sea, the Chla and temperature. However, other important parameters such as, temporal and spatial variability, salinity and nutrients seem to govern the plankton community metabolism within this particular ecosystem
and are not included in the abstract. Therefore, this lack of agreement between the ms, the consclusion and the abstract. is confusing. In my opinion, there is a large floor in the experimental design proposed and it is difficult to resolve.

*Reply*

We appreciate the reviewer's comment, and agree that the abstract highlighted our main findings and did not detail all the results. The abstract was indeed mostly orientated towards the effect of temperature and nutrient availability on metabolic rates as we found that those were the main controlling drivers. That was consistently explained in our results, discussion and conclusion; therefore, we do not find disagreement in our statements. However, following the suggestion of referee #1, we changed the narrative of the text to focus only on the variables highlighted in the abstract (i.e., temperature and nutrient availability; which are tightly related to the latitudinal gradient that characterises the basin) leaving aside the changes of planktonic metabolism throughout the water column.

*Action*

The changes can be tracked in the new version of the manuscript in the result section between lines 254–387. We have modified the figures accordingly so that Figure 2 now describes the latitudinal gradient of variables only in surface waters. In Figure 3 we have modified the format and the data presented, so now we only centre on data within the photic layer. We removed Figures 4–5, Figure 1A and 2A. We also removed Table 1, and Table 2. The information from Table 2 is now presented as a figure (now Figure 5) with the overall information and with all the data plotted. The changes in the results are in agreement with changes in the narrative of the discussion section. The main changes in the discussion section can be tracked between lines 447–463 and between lines 550–559

**Reviewer comment 2**
All samples included the deepest ones have been incubated on deck with surface water. During some of the surveys there is an important thermal variability. The authors have attempted to mitigate the issue by including just those samples above the thermocline. However, Material and Methods mention that changes in temperature and PAR in the incubation tanks were recorded with HOBO data loggers. Therefore, those data should be shown in a table in order to select objectively the samples for the analyses. Hence, eliminating those samples that register thermal differences above 2_C with the in situ temperatures. In addition, samples adapted to cool temperatures such as those at the bottom will respond more drastically to artificial increments of temperature than surface ones (for example. Apple et al. 2006. AME. 43: 243–254) resulting in erroneous conclusions. Therefore, Figure A1 is important and should be
included in the Ms.

*Reply*

As mentioned in our previous answer, after careful consideration, we decided to focused on the overall patterns observed in planktonic metabolism leaving aside the variability that takes place through the water column.

*Action*

We removed Figure A1 from the current version of the manuscript; however, all metabolic rates are now shown in the newly generated Figure 5.

**Reviewer comment 3**
Other figures such as 4-6 do not show crucial information in the current format. Figure 3 and Table 3 to me are redundant

*Action*

We removed Figures 4–6 from the current version of the manuscript, and the overall results are now presented as a new figure (Figure 5). We modified Table 3 (now Table 1) to only show complimentary information to Figure 3.

**Reviewer comment 4**
The paragraph 10-15 page 6 the authors should indicate if samples were collected before sunrise (to avoid any light on the samples) and if the incubation started at the sunrise to estimate the full light period. The authors say, the samples were colleted between 7 to 9 and to me this sounds very late to incubate and obtain the full light period nor precisely.

*Reply*

The samples were incubated for 24-h, covering an entire dark-light period, thus there is no need to estimate the light period.

**Reviewer comment 5**
In The net community metabolism..... page 7, NCP should be estimated during the light period (NCP 6 to 12 hours).

*Reply*

We believe that there is a misunderstanding regarding the process we measured. The reviewer's comment seemed to suggest that our work was focused on primary production, which is performed by the photosynthetic components of the plankton community during the daytime and that has gross and net components (as phytoplankton excrete and respire carbon). However, our paper focuses on the entire plankton community, both photosynthetic and heterotrophic (e.g. bacteria), where the net community production (NCP) represents the organic matter remaining after consumption of the GPP through respiration by plants (autotrophs), microbes (either autotrophs or heterotrophs), and animals (heterotrophs) (Ducklow and Doney 2013). Studies that focus on the photosynthetic component of the plankton community (e.g. Net Primary Production, NPP) report values for the daylight period only. Whereas studies, such as this, report (24 h) rates. For instance, published syntheses of community metabolism rates report values per day (24 h), e.g., Robinson and Williams 2005, Regaudie-de-Gioux 2012, 2013). The use of 24 h to report rates is justified as the metabolic budget needs be resolved over 24 h to be completed, not required for photosynthesis as it is light-dependent, but respiration, which is necessary to define net community production, which occurs both during day and at night.

**Reviewer comment 6**
The authors should show, the variation coefficient of the pool data and also the original CR, NCP and GPP data including their SE.

*Action*

The data are now shown in Figure 5, with a detailed description of the statistical analyses. In addition, the relevant information regarding the mean and the standard error of the mean are shown. See e.g., lines 351, 352, 359.

**Reviewer comment 7**
Because, In these oligrotrophic areas the metabolic rates are very low and can be difficult to detect. Therefore, the methodology needs to be very precise in the processesof filling, incubating and fixing the bottles.

*Reply*

We agree with the reviewer, however, the information about filling the bottles and all special cares during the sampling was detailed in methods section 2.3 (Page 7: lines 11–15). Now between lines 188–194.

**Reviewer comment 8**
The paragraph 20 in page 8 It should be indicated the Arrhenius plots the authors Mention

*Reply*

The Arrhenius plots were described in the methods section 2.3 (P8, between lines 13 and 20) and additionally explained in the results section 3.3 (P11: lines 16 and P12 lines 1–3) and Figures 9 and 10.

*Action*

The temperature-dependence of planktonic metabolism explained with the Arrhenius plots can now be seen between lines 379–387 and in Figures 7 and 8.

**Reviewer comment 9**
The paragraph 10 in page 10 should be transfered from the Results to the Discussion.

*Reply*

The sentence on P10, between lines 8–10 is a closing statement with the main results shown in previous paragraphs, we are not discussing any results. Therefore, we prefer to keep it as it is.

**Reviewers comment 10**
And also the first paragraph of the 3.2 Variability of plankton : : :. Is already mentioned in M and M.

*Reply*

We agree with the reviewer's comment and have modified the text accordingly

*Action*

We have deleted this sentence

**Reviewers comment 11**
The name of the KAUST is excessive. I would use just one larger map with different colours or shapes to show the stations at each survey or season.

*Reply*

We decided to not modify the figure as some of the stations were sampled in the same location more than twice, and different shapes or points overlap

*Action*

We did remove the name of the university to leave it only in one panel

**Reviewers comment 12**
Figure 2. I consider in this figure is difficult to detect the thermocline and the vertical profiles of Chla and salinity. I consider that nutrient profiles should also included

*Reply*

It was not possible to determine the depth of the thermocline in Figure 2, and it was not intended to do so. The figure summarises the main characteristics along the latitudinal axis that we sampled (i.e., the increasing temperature and phytoplankton chlorophyll-a toward the southern region with an increasing salinity towards the north).

*Action*

We modified Figure 2, so that now it only shows the variability of temperature, salinity and chlorophyll-a in surface waters.

**Reply to reviewer 13**
Figure 8, 9 and 10. To test one of the main conclusions, if AE is higher for GPP than for CR, authors should test statistically if the slopes are different. I would test also the slopes for the figures 9 and 10 explaining the consequences of the statistical differences in the cases observed

*Reply*

We performed an analysis of covariance (ANCOVA)

*Action*

We have mentioned this in section 2.4 between lines 249–251 and described the results between lines 382–387.

**Reply to reviewer 14**
In the figure 9, the RMA analyses have been included but it is not necesary in this case because temperature is not a rate. In addition, the authors have not explained when the RMA or OLS should be used in M and M.

*Reply*

We agree with the reviewer that it is not necessary to provide the results of the RMA analyses

*Action*

We removed these analyses and their results from our manuscript.

[revised manuscript text omitted]

---

## Author Response (AR3)

July 6, 2019

Dear Dr Ciavatta,
Associate Editor, Biogeosciences

We are re-submitting our manuscript "*Rates and drivers of Red Sea plankton community metabolism*" to be considered for publication in Biogeosciences. In this revised version of the manuscript, we have addressed all your comments and made changes where suggested, specifically:

1) We have modified the abstract (see lines 9–11) and consistently reported throughout the text the calculated activation energy as positive (e.g., see lines 17, 284, 286, 359). We added a sentence to the methods section to clarify why we use Ea values and not -Ea (obtained directly from the graphs) (please see lines 179-183).

2) We have specified the number of data included for the statistical analyses (e.g., lines 217–218, 226, 228) and consistently round the correlation coefficient values within the text and table 1.

3) Besides, we revised the manuscript for typos, and when needed, we improved specific sentences (e.g., line 42, 56–57).

4) We have included a new reference in line 84 (Lopez-Sandoval et al., 2019) as the study reported photosynthetic rates at different locations along the Red Sea.

Regarding the comment about statistical significance, we agree that a correlation does not mean causation or, as in our case, the lack of a significant relationship between metabolic rates and NOx does not imply that nutrients do no play a role regulating planktonic metabolism. Throughout the manuscript, we aimed to emphasise that not only temperature but also nutrient availability must be a significant driver regulating the metabolic response of planktonic communities in the Red Sea, as GPP and CR rates peaked in the warmer and more nutrient-enriched area of the basin.

We do not think that the lack of relationship between metabolic rates and NOx can be solely attributed to a reduced number of samples, which indeed was true (n=56 for NOx, n=77 for metabolic rates). In our results, we showed that NOx increased significantly from north to south in the surface layers and the bottom of the photic zone (Figure 3); however, this pattern was not evident when all data were taken in concert (data not shown). We also found that Chlorophyll-*a* concentration and metabolic rates were overall highest between these two layers, particularly in the southern Red Sea. Therefore, it seems conceivable to expect that there is an efficient consumption of nutrients occurring between these two layers, as NOx remained mostly low and constant between the surface and the depth receiving 1% of surface PAR along the basin. The likely fast turnover rate of the nutrients pools (a common feature in oligotrophic environments) fuels planktonic metabolism whenever nutrients are available. This rapid turnover of nutrients seems to be particularly accentuated in the southern region, hence, explaining the positive relationship found between metabolic rates and latitude. Therefore, it is possible that because nutrient concentration remained relatively similar within the area of higher productivity, we fail to see a correlation between metabolic rates and nutrient concentration.

We hope this explanation is satisfactory and that with the new changes you will find the revised version of the manuscript fulfils the quality and relevance necessary to be considered for publication in Biogeosciences.

Sincerely,

Daffne C. López-Sandoval

[revised manuscript text omitted]